# Validating the Lottery Ticket Hypothesis with Inertial Manifold Theory

**Zeru Zhang**[1*]  **Jiayin Jin**[1*]  **Zijie Zhang**[1]  **Yang Zhou**[1†]  **Xin Zhao**[1]
**Jiaxiang Ren**[1]  **Ji Liu**[2]  **Lingfei Wu**[3]  **Ruoming Jin**[4]  **Dejing Dou**[2,5]
[1]Auburn University, [2]Baidu Research, [3]JD.COM Silicon Valley Research Center,
[4]Kent State University, [5]University of Oregon
zzz0054@auburn.edu, jzj0042@auburn.edu, zzz0092@auburn.edu,
yangzhou@auburn.edu, cmkk684735@gmail.com, jzr0065@auburn.edu,
liuji04@baidu.com, lwu@email.wm.edu, rjin1@kent.edu,
dou@cs.uoregon.edu, doudejing@baidu.com

## Abstract

Despite achieving remarkable efficiency, traditional network pruning techniques often follow manually-crafted heuristics to generate pruned sparse networks. Such heuristic pruning strategies are hard to guarantee that the pruned networks achieve test accuracy comparable to the original dense ones. Recent works have empirically identified and verified the Lottery Ticket Hypothesis (LTH): a randomly-initialized dense neural network contains an extremely sparse subnetwork, which can be trained to achieve similar accuracy to the former. Due to the lack of theoretical evidence, they often need to run multiple rounds of expensive training and pruning over the original large networks to discover the sparse subnetworks with low accuracy loss. By leveraging dynamical systems theory and inertial manifold theory, this work theoretically verifies the validity of the LTH. We explore the possibility of theoretically lossless pruning as well as one-time pruning, compared with existing neural network pruning and LTH techniques. We reformulate the neural network optimization problem as a gradient dynamical system and reduce this high-dimensional system onto inertial manifolds to obtain a low-dimensional system regarding pruned subnetworks. We demonstrate the precondition and existence of pruned subnetworks and prune the original networks in terms of the gap in their spectrum that make the subnetworks have the smallest dimensions.

## 1 Introduction

Deep learning techniques utilize the architecture of large model size and over-parameterization to achieve state-of-the-art performance in a wide range of applications [17, 18]. The performance improvements can be owed to the availability of large-scale neural networks and massive computational resources to train such complex models. However, large model size and extreme over-parameterization typically lead to high training cost, slow inference speed, and large memory consumption. This hinders the applicability of deep learning models in resource-intensive scenarios with the requirement of low latency and energy consumption, such as Internet of Things [2] and mobile computing [51, 55].

We have witnessed a large number of research advances in neural network pruning for reducing the time and space requirements of deep neural networks both at training and test time, with minimal loss in accuracy [31, 20, 34]. These techniques identify non-essential weights, filters, and other structures

---

*Equal contributors

†Corresponding author

35th Conference on Neural Information Processing Systems (NeurIPS 2021).

from neural networks that do not contribute significantly to the model performance, and remove them for compressing original dense neural networks [49, 68, 52, 57, 3, 30, 35, 64, 39, 27, 29, 4, 40, 61, 33, 56, 16, 21]. The goal of neural network pruning is to reduce computational cost and memory consumption without a significant accuracy drop, compared with the original dense networks.

However, the majority of neural network pruning methods require to train a dense neural network to generate a pruned sparse network, as it is difficult to train a pruned network from scratch [20, 62, 34, 23, 41, 67, 49, 68, 52, 57, 3, 30, 74, 24, 25, 63, 35, 64, 39, 27, 29, 4, 40], although there are few pruning before training approaches [32, 61, 33, 56, 16, 21]. Thus, these methods still suffer from the non-trivial training cost on over-parameterized networks. Most importantly, traditional network pruning techniques are based on ad-hoc heuristic pruning strategies [20, 74, 63, 29], leaving some critical theoretical questions unanswered on how much we can prune a neural network given a specified tolerance of accuracy drop, and how to achieve it with a practical and efficient procedure. Therefore, it is difficult to theoretically guarantee that the pruned networks achieve test accuracy comparable to the original dense ones. In addition, a recent literature has demonstrated that traditional neural network pruning often fall-short compared to the Lottery Ticket Hypothesis techniques [33].

A recent report has empirically identified and validated a surprising finding of the Lottery Ticket Hypothesis (LTH): a randomly-initialized dense neural network contains an extremely sparse subnetwork (i.e., a winning lottery ticket) such that, when trained from scratch with weights being reset to its initialization, can achieve similar performance to the original dense network within similar training iterations [13, 14]. Compared with the original dense networks, the winning tickets are able to save a lot of memory for retraining [13, 73, 14, 15, 70, 42, 53, 11, 8]. The existence of winning tickets indicates that deep learning models can be trained in low-resource environments.

While the LTH techniques exhibit promising practical implications for scaling over large models and datasets, they fail to provide an efficient algorithm to search for the winning ticket subnetworks for deep learning models in practice, due to the lack of theoretical evidence and analysis. They often utilize an Iterative Magnitude Pruning (IMP) strategy to gradually find winning tickets, which needs multiple rounds of expensive training, pruning, and resetting over large neural networks. Although existing LTH techniques raise very intriguing observations, most of them provide only empirical evidence to verify the LTH [71, 12, 1, 47, 69, 54, 5, 53, 26, 8, 7, 11]. In this case, it is difficult to identify the optimal pruning strategy without performance drop, i.e., how and where a neural network should be pruned. Therefore, two fundamental unanswered questions are raised: (1) How to verify the existence of winning lottery tickets, i.e., sparse trainable subnetworks inside a dense neural network, with theoretical evidence and analysis? (2) How to provide efficient algorithms to discover winning lottery tickets, without multiple rounds of expensive training and pruning in the IMP process?

To our best knowledge, this work is the first to theoretically verify the LTH and the existence of winning lottery tickets by leveraging dynamical systems theory and inertial manifold theory.

We reformulate gradient descent optimization of a neural network as a gradient dynamical system regarding $W$ (i.e., parameters of the neural network) and analyze its dynamics with dynamical systems theory. Since inertial manifolds $\mathcal{M}_{\mathcal{W}}$ contain all local minimum points $W^*$ of loss function $L$ regarding $W$ (i.e., $\nabla L(W^*) = 0$), we reduce the original high-dimensional gradient system $\varphi$ onto inertial manifolds to obtain a low-dimensional system regarding $W^+$, i.e., parameters of the winning ticket subnetwork. We demonstrate that there must exist such $\mathcal{M}_{\mathcal{W}}$ for neural network pruning, when the gradient dynamical system meets a certain condition about its spectrum and Lipschitz constant, for cutting $W$ into $W^+$ and $W^-$ (i.e., redundant parameters of the original neural network).

In comparison with existing neural network pruning and LTH techniques, our dynamical systems and inertial manifold-based method exhibits two unique advantages: (1) All local minimum points $W^*$ lie in $\mathcal{M}_{\mathcal{W}}$ and thus $W^+$ is theoretically lossless pruning of $W$. (2) The computation of $\mathcal{M}_{\mathcal{W}}$ and $W^+$ is a one-time operation, such that our method is able to avoid the cost of multiple rounds of expensive training and pruning on large networks in the LTH methods. Due to large size of $W$ in real scenarios, an approximate method is designed to estimate the Lipschitz constant for maintaining the efficiency.

Extensive evaluation on real datasets demonstrates the superior performance of our proposed IMC model against several state-of-the-art neural network pruning and LTH methods. More experiments, implementation details, and hyperparameter setting are presented in Appendices A.3-A.5.

## 2 Background

### 2.1 Lottery Ticket Hypothesis

The Lottery Ticket Hypothesis (LTH) states that *a randomly-initialized, dense neural network contains a subnetwork that is initialized such that, when trained in isolation, it can match the test accuracy of the original network after training for at most the same number of iterations* [13, 14].

Given a dense neural network $f(x; W)$ with randomly-initialized weight parameters $W = W_0 \in \mathbb{R}^d$, when optimizing with stochastic gradient descent (SGD) on a training set, $f$ reaches minimum loss $L$ at iteration $I$ with test accuracy $a$. A sparse subnetwork of $f$ is denoted as $f(x; M \odot W)$ using a binary pruning mask $M \in \{0, 1\}^d$, $\|M\|_0 \ll d$ on its parameters such that its initialization is $M \odot W_i$, where $\odot$ denotes the Hadamard product and $W_i$ are the generated parameters of the original model $f(x; W)$ at iteration $i$. When optimizing with SGD on the same training set, $f(x; M \odot W)$, where $W$ is initialized with $W_i$, can be trained from $M \odot W$ to reach minimum loss $L$ at commensurate iteration $\bar{I}$ with commensurate accuracy $\bar{a} \geq a$ to the original model $f(x; W)$.

The LTH techniques often utilize an Iterative Magnitude Pruning (IMP) strategy to find winning tickets by gradually training the network and pruning its smallest-magnitude weights [13–15]. It operates in $T$ rounds of expensive training, pruning, and resetting over large neural networks for guaranteeing the acceptable accuracy drop. Each of multiple rounds in the IMP consists of four steps: (1) Randomly initialize a dense neural network $f(x; W)$ with weight parameters $W = W_0$; (2) Train $W$ of $f$ for $I$ iterations, generating updated parameters $W_I$; (3) Prune $p\%$ of the parameters with the smallest magnitude in $W_I$, producing a mask $M$; and (4) Reset the remaining weight parameters back to their initial values in $W_0$, creating the winning ticket $f(x; M \odot W_0)$.

### 2.2 Dynamical Systems Theory

In mathematics, an ordinary differential equation (ODE) is a differential equation containing one or more functions of one independent variable and the derivatives of those functions [75]. A system of first-order ODE is defined as follows.

$$x'(t) = f(x(t)), \ x \in X \tag{1}$$

where $X$ is a Banach space and $f : X \to X$ is a function of $x(t)$. The unknown function $x(t)$ at time $t$ appears on both sides of the differential equation.

**Definition 1.** *[Globally Lipschitz] $f(x)$ is globally Lipschitz in $X$ if there exists a constant $K \geq 0$ for all $x, y \in X$, it holds $\|f(x) - f(y)\|_X \leq K\|x - y\|_X$, where $K$ is the Lipschitz constant of $f$.*

**Definition 2.** *[Global Solution of ODE] A solution $x(t) : \mathbb{R} \to X$ of the ODE in Eq.(1) is said to be global if $x(t)$ exists for any $t \in \mathbb{R}$.*

**Lemma 1.** *If $f(x)$ is globally Lipschitz in $X$, then the ODE in Eq.(1) has a unique global solution in $X$ for any initial condition $x(0) = x_0$ [9].*

**Definition 3.** *[Flow of ODE] $\varphi : \mathbb{R} \times X \to X$, defined by $\varphi(t, x_0) = x(t)$ is the flow (i.e., solution map) of the ODE in Eq.(1) if $\varphi(t, x_0)$ is the unique solution of the ODE with initial condition $\varphi(0, x_0) = x(0) = x_0$. For any given initial value $x_0$, the sets $\{\varphi(t, x_0) : t \in \mathbb{R}\}$, $\{\varphi(t, x_0) : t \in \mathbb{R}^+\}$, $\{\varphi(t, x_0) : t \in \mathbb{R}^-\}$ are called the complete orbit, positive orbit, and negative orbit starting from $x_0$, respectively.*

In this work, we only consider the case of positive time and use the positive orbit starting from $x$. The orbit $\{\varphi(t, x_0) : t \in \{\mathbb{R}^+ \cup 0\}\}$ of the flow is also called the trajectory of the point $x$ under the flow.

**Definition 4.** *[Continuous Dynamical System] If a flow $\varphi(t, x_0)$ satisfies the following properties:*

- *$\varphi(s + t, x_0) = \varphi(s, x_0)\varphi(t, x_0)$ for any $s, t \in \mathbb{R}$;*
- *$\varphi(t, x_0)$ is continuous in $x_0$ for any $t \in \mathbb{R}$;*
- *$\varphi(t, x_0)$ is continuous in $t$ for any $x_0 \in X$,*

*then the flow $\varphi(t, x_0)$ together with the space $X$ is a continuous dynamical system.*

If $f(x)$ is globally Lipschitz, then the flow of the ODE is a continuous dynamical system. Dynamical system is a mathematical concept that is used to describe the time dependence of a point's position in its ambient space. Dynamical System Theory is a mathematical tool to capture long-time behavior of solutions.

**Definition 5.** *[Equilibrium of Dynamical System] A point $x^*$ is called an equilibrium of a dynamical system if $\varphi(t, x^*) = x^*$ for all $t \in \mathbb{R}$, i.e., $f(x^*) = 0$.*

An equilibrium is a solution that does not change with time. This means if the systems starts at an equilibrium, the state will remain at the equilibrium forever.

**Definition 6.** *[Invariant Set/Manifold] A set or manifold $A$ is invariant under a dynamical system $\varphi$ if $\varphi(t, A) \subset A$ for any $t \geq 0$. That is, if $a$ is an element of $A$ then so is $\varphi(t, a)$ for all $t \geq 0$.*

A manifold is a topological space that locally resembles Euclidean space near each point. Examples of invariant manifolds include stable manifold, unstable manifold, and inertial manifold.

**Definition 7.** *[Attractor of Dynamical System] A set $A \subset \mathbb{R}^n$ is an attractor that attracts a set $B \subset \mathbb{R}^n$ under a dynamical system $\varphi$ if $A$ is an invariant set under $\varphi$ and*

$$\lim_{t \to \infty} \text{dist}(\varphi(t, B), A) = 0 \tag{2}$$

*where $B$ denotes the basin of attraction for $A$. dist represents the Hausdorff semi-distance defined by*

$$\text{dist}(B, A) = \sup_{x \in B} \inf_{y \in A} \|x - y\|_X \tag{3}$$

An attractor is a set of states (i.e., points in the phase space), invariant under the dynamics, towards which neighboring states in a given basin of attraction asymptotically approach in the course of dynamic evolution. An attractor can be a point, a finite set of points, a curve, a manifold, or even a complicated set with a fractal structure. $\lim_{t \to \infty} \text{dist}(\varphi(t, B), A) = 0$ implies that flows starting from any point $b \in B$ enter $A$ in the limit $t \to \infty$.

**Definition 8.** *[Global Attractor of Dynamical System] A nonempty compact set $A \subset X$ is a global attractor under a dynamical system $\varphi$ if $A$ is an invariant set under $\varphi$ and $A$ attracts all bounded sets $B \subset X$.*

A global attractor represents the maximal compact invariant set which attracts the orbits of all bounded sets at a uniform rate.

**Definition 9.** *[Stable/Unstable Manifold of Dynamical System] Given an equilibrium $x^*$ of a dynamical system, the stable manifold $\mathcal{M}^s(x^*)$ of $x^*$ is a set of points in phase space if*

$$\mathcal{M}^s(x^*) = \{x \in X : \varphi(t, x) \to x^* \text{ as } t \to +\infty\} \tag{4}$$

*and the unstable manifold $\mathcal{M}^u(x^*)$ of $x^*$ is a set of points in phase space if*

$$\mathcal{M}^u(x^*) = \{x \in X : \varphi(t, x) \to x^* \text{ as } t \to -\infty\} \tag{5}$$

## 3 Validating the Lottery Ticket Hypothesis

The idea of this work is to reformulate gradient descent optimization of a neural network in Eq.(6) as a gradient system in Eq.(7) regarding $W$. By leveraging the dynamical system theory, we analyze the dynamics and equilibria (i.e., local minimum points $W^*$ of loss function $L$ in Eq.(6)) of the gradient dynamical system and theoretically validate the LTH. The gradient dynamical system has a global attractor that contains all equilibria. However, the global attractor usually has fractal dimension and its geometry is very complicated. Thus, it is difficult to analyze the dynamics inside it.

Fortunately, inertial manifolds $\mathcal{M}_W$ of the gradient system enclose the global attractor and have manifold structure (i.e., regular geometric objects). It is more convenient to analyze the dynamics inside them and reduce the original high-dimensional gradient system regarding $W$ onto $\mathcal{M}_W$ to obtain a low-dimensional system regarding $W^+$, i.e., parameters of the winning ticket subnetwork. According to Theorem 5, the new low-dimensional system governs the original high-dimensional one. Most importantly, it encloses all equilibria of the original system, i.e., all $W^*$ lie in the inertial manifold. Thus, it is efficient to directly analyze the low-dimensional system to discover the $W^*$.

### 3.1 Neural Network Pruning

Traditional deep learning models often utilize gradient descent techniques to optimize neural networks and reach a local minimum of loss $L$. The gradient decent procedure is given below.

$$W(t + 1) = W(t) - \eta \nabla L(W(t)) \tag{6}$$

where $W$ are the parameters of neural network, $t$ is the iteration time, $\eta$ is a learning rate, and $L$ is the loss function. By starting from initial parameters $W(0)$, the gradient decent procedure converges to a local minimum point $W^*$ of $L$, i.e., $\nabla L(W^*) = 0$.

The above gradient decent procedure in Eq.(6) can be viewed as the numerical solution of an ODE system under classical Euler scheme. The sequence of $W(0), W(1), \cdots, W^*$ in the gradient descent process in Eq.(6) corresponds to discrete points on the orbit of the following ODE.

$$W'(t) = -\nabla L(W(t)) \tag{7}$$

$W^*$ is an equilibrium (i.e., a time-independent solution) of the ODE system. The ODE in Eq.(7) is a gradient system, since the vector field $-\nabla L(W)$ is the gradient of $L$. The dynamical system derived by (7) is a gradient dynamical system, where the loss is decreasing along the solution of the equation.

$$\frac{d}{dt} L(W(t)) = \nabla L \cdot \frac{dW(t)}{dt} = -|\nabla L(W(t))|^2 \le 0 \tag{8}$$

This is the reason why the gradient descent method works for the optimization of neural networks.

The following theorems analyze the existence, uniqueness, and convergence of the global attractor $A$ of the gradient dynamical system derived by (7). Most importantly, they demonstrate that all local minimum points $W^*$ of $L$ of the original neural network in Eq.(6) lie in $A$ of the system by (7).

**Theorem 1.** *The global attractor $A$ of the dynamical system derived by Eq.(7) is the union of equilibria $W^*$ and their unstable manifolds $\mathcal{M}^u(W^*)$ [19].*

Since the equilibria $W^*$ are the time-independent solutions of our gradient dynamical system, they correspond to the local minimum points of $L$ of the original neural network in Eq.(6).

**Theorem 2.** *The global attractor $A$ of the dynamical system derived by Eq.(7) is unique.*

**Theorem 3.** *Let $A \subset \mathcal{W}$ be a compact subset of a Banach space $\mathcal{W}$, and $W(t) \in \mathcal{W}$ be a sequence with $\lim_{t\to\infty} \mathrm{dist}(W(t), A) = 0$, then $W(t)$ has a convergent subsequence, whose limit lies in $A$.*

**Theorem 4.** *The global attractor $A$ of the dynamical system derived by Eq.(7) is the maximal compact invariant set, and the minimal set that attracts all bounded sets $B \subset \mathcal{W}$.*

*Please refer to Appendix A.2 for detailed proof of Theorems 1-3.*

In mathematics, inertial manifolds are concerned with the long-term behavior of the solutions of dynamical systems [58]. Inertial manifolds are finite-dimensional, smooth, invariant manifolds that contain the global attractor and attract all solutions of the dynamical systems. Since an inertial manifold is lower-dimensional compared with the whole solution space and all important dynamics take place on inertial manifold, studying the dynamics on inertial manifolds only is enough to derive the dynamics of the original gradient system.

**Definition 10.** *[Inertial Manifold of Dynamical System] Let $\mathcal{W}$ be a Banach space, an inertial manifold $\mathcal{M}_W$ for a dynamical system is a smooth manifold such that*

- *$\mathcal{M}_W$ is of finite dimension;*
- *The flow $\varphi(t, \mathcal{M}_W) \subset \mathcal{M}_W$ for any $t \ge 0$;*
- *There exist two constants $C_1, C_2 > 0$, such that for any initial condition $W(0) \in \mathcal{W}$, it holds*

$$\mathrm{dist}(\varphi(t, W(0)), \mathcal{M}_W) \le C_1 e^{-C_2 t} \mathrm{dist}(W(0), \mathcal{M}_W) \tag{9}$$

Since all solutions of Eq.(6) eventually converge to the inertial manifold $\mathcal{M}_W$ and all local minimum points $W^*$ of $L$ of the original neural network in Eq.(6) locate on $\mathcal{M}_W$, it would be sufficient to work on the inertial manifolds $\mathcal{M}_W$ directly for discovering $W^*$.

**Theorem 5.** *Let $\mathcal{W}$ be a Banach space with dimension $d$, given a semi-linear ODE*

$$W'(t) = AW(t) + f(W(t)), \ W(t) \in \mathcal{W} \tag{10}$$

*where $A$ is a $d \times d$ matrix and $f : \mathcal{W} \to \mathcal{W}$. The spectrum of $A$ is denoted as follows.*

$$\lambda_1 \le \lambda_2 \le \cdots \le \lambda_d \tag{11}$$

*Assuming that $f(0) = 0$, let $K$ be the Lipschitz constant of $f(x)$, that is,*

$$\|f(x) - f(y)\|_\mathcal{W} \le K\|x - y\|_\mathcal{W} \text{ for any } x, y \in \mathcal{W} \tag{12}$$

*If there exits an integer $k$, such that $\lambda_{k+1} - \lambda_k > 4K$, let $\mathcal{W}^+$ be the generalized eigenspace for $\lambda_{k+1}, \ldots, \lambda_d$ and $\mathcal{W}^-$ be the generalized eigenspace for $\lambda_1, \ldots, \lambda_k$, then there exists an inertial manifold $\mathcal{M}_\mathcal{W}$ as follows.*

$$\mathcal{M}_\mathcal{W} = (\mathcal{W}^+, h(\mathcal{W}^+)) \tag{13}$$

*where $h$ is a mapping from $\mathcal{W}^+$ to $\mathcal{W}^-$.*

*Please refer to Appendix A.2 for detailed proof of Theorem 5.*

Theorem 5 demonstrates the whole solution space $\mathcal{W}$ can be decomposed into two subspaces $\mathcal{W}^+$ and $\mathcal{W}^-$ when the gap between arbitrary two consecutive eigenvalues in the spectrum of $A$ is larger than $4K$. An inertial manifold $\mathcal{M}_{\mathcal{W}}$ can be constructed as the graph of a mapping $h : \mathcal{W}^+ \to \mathcal{W}^-$. Consequently, the dynamics on $\mathcal{M}_{\mathcal{W}}$ can be described by the projected dynamics in $\mathcal{W}^+$ through the mapping $h$, and $\mathcal{W}^-$ is redundant. When there are multiple integers $k$ satisfying $\lambda_{k+1} - \lambda_k > 4K$, we choose the largest $k$ to cut $\mathcal{M}_{\mathcal{W}}$, such that $\mathcal{W}^+$ has the smallest dimensions, i.e., the best pruning.

In order to utilize the conclusion of Theorem 5 to prune the network parameters $W$ in Eq.(6), we leverage linearization-around-fixed-points technique in dynamical systems theory to transform the gradient system in Eq.(7) into an equivalent semi-linear ODE system as follows.

$$U(t) = W(t) - W^* \tag{14}$$

For ease of presentation, we use symbols $U$ and $W$ to replace $U(t)$ and $W(t)$. Then we have

$$U' = -H(W^*)U - \big(\nabla L(U + W^*) - H(W^*)U\big) = -H(W^*)U + F(U) \tag{15}$$

where $H(W^*)$ denotes the Hessian matrix of loss function $L$ evaluating at $W^*$ and $F(U)$ is equal to $-\big(\nabla L(U + W^*) - H(W^*)U\big)$.

The advantage of such linearization-around-fixed-points technique is to make the nonlinear term $F(U)$ satisfy $F(0) = 0$ since $\nabla L(W^*) = 0$. Notice that the assumption for nonlinear term $f(W(t))$ in Theorem 5 is $f(0) = 0$. The above linearization-around-fixed-points technique allows to leverage the conclusion of Theorem 5 for lossless neural network pruning.

Compared with the system about original parameters $W$ in Eq.(10), Eq.(15) is a semi-linear ODE system regarding converted parameters $U$. The corresponding inertial manifold $\mathcal{M}_U$ is given below.

$$\{\mathcal{M}_{\mathcal{U}} = (U^+, h(U^+)) : U^+ \in \mathcal{U}^+\} \tag{16}$$

where $\mathcal{U}^+$ is a corresponding subspace. $\mathcal{M}_{\mathcal{U}}$ is an invariant manifold, that is, for any solution $U(t)$ of the ODE system, if $U(0) \in \mathcal{M}_{\mathcal{U}}$, then $U(t) \in \mathcal{M}_{\mathcal{U}}$ for any $t \geq 0$. Moreover, $\mathcal{M}_{\mathcal{U}}$ contains all local minimum points. $U^+$ is a lossless pruning of $U$ according to the conclusion of Theorem 5.

Thus, the corresponding inertial manifold $\mathcal{M}_{\mathcal{W}}$ for the gradient system in Eq.(7) is given as follows.

$$\mathcal{M}_{\mathcal{W}} = W^* + \mathcal{M}_{\mathcal{U}} = \{(W^+, W^-) : U^+ \in \mathcal{U}^+\} \tag{17}$$

where $W^+ = U^+ + W^{*+}$ and $W^- = h(U^+) + W^{*-}$ and $W^{*+}$. $W^{*-}$ are the counterparts of $W^*$ in terms of the same dimension cutting as $U$. $W^*$ is a constant when we are given a trained large neural network. $W^-$ can be written as a function of $W^+$: $W^- = h(U^+) + W^{*-} = h(W^+ - W^{*+}) + W^{*-}$. We denote this function as $g$, i.e., $W^- = g(W^+)$.

The dynamics of the gradient system in Eq.(7) is governed by $\mathcal{M}_{\mathcal{W}} = (W^+, g(W^+))$. The system in Eq.(7) can be reduced onto $\mathcal{M}_{\mathcal{W}}$. The neural network and the loss function on $\mathcal{M}_{\mathcal{W}}$ are given below.

$$\bar{f}(W^+) = f\big(W^+, g(W^+)\big) = f(W), \quad \bar{L}(W^+) = L\big(W^+, g(W^+)\big) = L(W) \tag{18}$$

where $f$ and $\bar{f}$ represent the original and pruned neural networks respectively. The reduced gradient system on $\mathcal{M}_W$ is given as follows.

$$\frac{dW^+}{dt} = -\nabla \bar{L}(W^+) = -\nabla_{W^+} L\big(W^+, g(W^+)\big) - \nabla_{W^-} L\big(W^+, g(W^+)\big)\nabla g(W^+) \tag{19}$$

As all local minimum points lie in $\mathcal{M}_W$, the reduced problem in Eqs.(18) and (19) work equally well as the original one in Eq.(7). This indicates that using the low-dimensional parameters $W^+ \in \mathbb{R}^{d-k}$ to train $\bar{f}$ is sufficient to obtain the same performance as training $f$ with the original high-dimensional parameters $W \in \mathbb{R}^d$. Therefore, we only need to train a subnetwork with $W^+$ as parameters.

In order to further improve the pruning efficiency, we train the parameters $W$ of the original networks few iterations (10 epochs in our implementation) to generate an approximate $W^*$. Due to large size of $W^*$, we employ a scalable Hessian computation method to calculate an approximate $H(W^*)$ [65, 66]. We prune the original network with parameters $W$ to generate a subnetwork with parameters $W^+$ and train the latter until convergence.

## 3.2 Estimation of Lipschitz Constant

In order to leverage the conclusion of Theorem 5 to compress the parameters $W$ of neural networks in Eq.(6), we need to verify whether its condition is satisfied and thus we need to estimate the Lipschitz constant of the nonlinear term $F(U)$ in Eq.(15).

Let $B$ be the $l_\infty$-upper bound of $U$, i.e., $\|U\|_\infty \leq B$, by utilizing the Mean Value Theorem, we get

$$
\begin{aligned}
F(U_1) - F(U_2) &= \nabla L(U_1 + W^*) - \nabla L(U_2 + W^*) - H(W^*)(U_1 - U_2) \\
&= \left[H\left(W^* + U_1 + \theta_1(U_2 - U_1)\right) - H(W^*)\right](U_1 - U_2) \\
&= (U_1 - U_2)^T \nabla H\left(W^* + U_1 + \theta_2(U_2 - U_1)\right)(U_1 - U_2)
\end{aligned}
\tag{20}
$$

where $0 \leq \theta_1, \theta_2 \leq 1$ are two constants.

By employing the Hölder Inequality, we have

$$
\|F(U_1) - F(U_2)\|_p \leq \|\nabla H\left(W^* + U_1 + \theta_2(U_2 - U_1)\right)\|_p \|U_1 - U_2\|_\infty^2, \ 1 \leq p \leq +\infty
\tag{21}
$$

As $\|U\|_\infty \leq B$, $\|U_1 - U_2\|_\infty \leq 2B$. In addition, $\|U_1 - U_2\|_\infty \leq \|U_1 - U_2\|_p$. Thus, we have

$$
\|F(U_1) - F(U_2)\|_p \leq 2B\|\nabla H\left(W^* + U_1 + \theta_2(U_2 - U_1)\right)\|_p \|U_1 - U_2\|_p
\tag{22}
$$

Therefore, the Lipschitz constant of $F(U)$ in $l_p$ space is $2B\|\nabla H\left(W^* + U_1 + \theta_2(U_2 - U_1)\right)\|_p$.

For ease of presentation, we use a symbol $V$ to replace $W^* + U_1 + \theta_2(U_2 - U_1)$. Since $2B$ is a constant for a given $U$, we only need to estimate $\|\nabla H(V)\|_p$.

Based on the Second Derivative Formula, the estimation of $\|\nabla H(V)\|_p$ is given as follows.

$$
\|\nabla H(V)\|_p \approx \left\|\frac{\nabla L(V + \Delta M/\|M\|_p) - 2\nabla L(V) + \nabla L(V - \Delta M/\|M\|_p)}{\Delta^2}\right\|_p
\tag{23}
$$

where $\Delta$ refers to corresponding change of $V$.

In our implementation, we choose $p = 2$ and $\Delta = 0.01$. $A$ is a randomly-generated matrix with the same dimension as $V$. In the current experiments, the estimated Lipschitz constants of all neural networks are very tiny with order $10^{-12}$. This implies that our neural network pruning technique has great potential as a general pruning solution to other neural networks, which is desirable in practice.

## 3.3 Algorithm

The following are the algorithm descriptions of our Inertial Manifold-based neural network Compression (IMC) method step by step: (1) Given a dense neural network $f(x; W)$ with randomly-initialized flattened weight parameters $W = W_0 \in \mathbb{R}^d$, when optimizing $W$ with stochastic gradient descent (SGD) on a training set, we generate an approximate $W^*$ in few training iterations (10 iterations in our implementation), where $W^*$ is a local minimum point of loss function $L$ regarding $W$ in Eq.(6) (i.e., $\nabla L(W^*) = 0$). Notice that $W^*$ is also an equilibrium of the gradient dynamical system in Eq.(7); (2) Based on the transformation $U = W - W^*$ from $W$ to $U$ through the above approximate $W^*$ in Eq.(14), we transform the gradient dynamical system in Eq.(7) into a semi-linear ODE system $U' = -H(W^*)U + F(U)$ in Eq.(15), where $H(W^*)$ denotes the Hessian matrix of loss function $L$ evaluating at $W^*$ and $F(U)$ is equal to $-\left(\nabla L(U + W^*) - H(W^*)U\right)$; (3) We employ a scalable Hessian computation method to calculate an approximate $-H(W^*)$ [78, 79]. We calculate the spectrum of $-H(W^*)$ and sort its eigenvalues as $\lambda_1 \leq \lambda_2 \leq \cdots \leq \lambda_d$; (4) We estimate the Lipschitz constant $K$ of the nonlinear term $F(U)$ in Eq.(15) based on Eqs.(22) and (23); (5) We check pairwise consecutive eigenvalues in descending order, i.e., $\lambda_d$ and $\lambda_{d-1}$, $\lambda_{d-1}$ and $\lambda_{d-2}$, $\cdots$, and $\lambda_2$ and $\lambda_1$. When finding the first $k$ satisfying $\lambda_{k+1} - \lambda_k > 4K$, we partition $U \in \mathbb{R}^d$ into $U^+ \in \mathbb{R}^{d-k}$ and $U^- \in \mathbb{R}^k$ and split $W \in \mathbb{R}^d$ into $W^+ \in \mathbb{R}^{d-k}$ and $W^- \in \mathbb{R}^k$; and (6) We prune the original network with high-dimensional parameters $W \in \mathbb{R}^d$ to generate a subnetwork with low-dimensional parameters $W^+ \in \mathbb{R}^{d-k}$, i.e., reduce $W$ to $W^+$, and train $W^+$ until convergence.

# 4 Experimental Evaluation

In this section, we have evaluated the effectiveness of our IMC model and other baselines for neural network pruning over three standard image classification datasets: CIFAR-10 [28], CIFAR-100 [28],

Table 1: Accuracy with found sparsity by our IMC method on CIFAR-10

| Neural Network | ResNet-20 | | | | ResNet-32 | | | |
|---|---|---|---|---|---|---|---|---|
| Metric | Sparsity | Accuracy | Epoch | Runtime (s) | Sparsity | Accuracy | Epoch | Runtime (s) |
| Baseline | 0 | 94.6 | 160 | 1,956 | 0 | 95.0 | 160 | 4,233 |
| Flow&Prune | 0.47 | 91.6 | 160 | 6,120 | 0.45 | 92.6 | 160 | 8460 |
| SNIP | 0.47 | 92.4 | 160 | 2,675 | 0.45 | 92.9 | 160 | 4,046 |
| SynFlow | 0.47 | 92.5 | 160 | 2,664 | 0.45 | 90.8 | 160 | 4,046 |
| LTH+Reinitialization | 0.47 | 92.9 | 1,120 | 24,901 | 0.47 | 93.6 | 1,120 | 100,301 |
| LTH+Rewinding | 0.47 | 92.5 | 1,120 | 24,477 | 0.47 | 93.6 | 1,120 | 100,328 |
| LTH+FineTuning | 0.47 | 91.5 | 1,120 | 24,875 | 0.47 | 92.8 | 1,120 | 100,291 |
| GraSP | 0.47 | 91.1 | 160 | 2,966 | 0.45 | 91.5 | 160 | 4,659 |
| sanity-check | 0.47 | 93.3 | 160 | **2,185** | 0.45 | 93.5 | 160 | **3,009** |
| Continuous Sparsification | 0.47 | 92.2 | 320 | 5,298 | 0.44 | 92.9 | 320 | 7,490 |
| IMC-Reinitialization | 0.47 | 94.5 | 160 | 3,215 | 0.45 | 94.8 | 160 | 4,704 |
| IMC-Rewinding | 0.47 | 94.5 | 160 | 3,179 | 0.45 | 94.9 | 160 | 4,591 |
| IMC-FineTuning | 0.47 | **94.6** | 160 | 3,205 | 0.45 | **95.1** | 160 | 4,607 |

Table 2: Sparsity with highest accuracy on CIFAR-10

| Neural Network | ResNet-20 | | | | ResNet-32 | | | |
|---|---|---|---|---|---|---|---|---|
| Metric | Sparsity | Accuracy | Epoch | Runtime (s) | Sparsity | Accuracy | Epoch | Runtime (s) |
| Baseline | 0 | 94.6 | 160 | 1,956 | 0 | 95.0 | 160 | 4,233 |
| Flow&Prune | 0.1 | 94.1 | 160 | 6,124 | 0.1 | 94.5 | 160 | 8,371 |
| SNIP | 0.1 | 92.9 | 160 | 2,670 | 0.1 | 93.4 | 160 | 4,075 |
| SynFlow | 0.05 | 93.1 | 160 | 2,659 | 0.15 | 93.6 | 160 | 4,065 |
| LTH+Reinitialization | 0.1 | 94.5 | 320 | 6,940 | 0.19 | 94.9 | 480 | 44,636 |
| LTH+Rewinding | 0.1 | 94.5 | 320 | 7,060 | 0.19 | 94.1 | 480 | 44,683 |
| LTH+FineTuning | 0.1 | 94.2 | 320 | 7,101 | 0.1 | 94.5 | 320 | 30,406 |
| GraSP | 0.05 | 92.8 | 160 | 3,081 | 0.05 | 93.4 | 160 | 4,637 |
| sanity-check | 0.1 | 94.2 | 160 | 2,124 | 0.15 | 94.5 | 160 | 3,138 |
| Continuous Sparsification | 0.39 | 92.5 | 320 | 5,261 | 0.25 | 93.3 | 320 | 7,067 |
| IMC-Reinitialization | **0.47** | 94.5 | 160 | 3,215 | **0.45** | 94.8 | 160 | 4,704 |
| IMC-Rewinding | **0.47** | 94.4 | 160 | 3,179 | **0.45** | 94.9 | 160 | 4,591 |
| IMC-FineTuning | **0.47** | 94.6 | 160 | 3,205 | **0.45** | 95.1 | 160 | 4,607 |

and ImageNet [10]. The experiments exactly follow the same settings described by the original LTH paper [13, 14] and other following works on LTH and network pruning [70, 59, 15, 53, 69, 11].

**Baselines.** We compare the IMC model with nine state-of-the-art models, including three network pruning and six LTH approaches. **Flow&Prune** is a pruning during training approach that uses gradient flow to determine the relationship between pruning measures and evolution of model parameters [40]. **SNIP** and **SynFlow** prune a given network once at initialization prior to training [32, 56]. **LTH+Reinitialization** is the original LTH method, which randomly sample a new initialization for winning tickets' initialization [13]. **LTH+Rewinding** rewinds unpruned weights to their values from earlier in training and retrains them from there using the original training schedule [15]. **LTH+FineTuning** fine-tunes the pruned model to regain the lost performance, instead of training it from scratch [38]. **GraSP** prunes the weights whose removal will result in least decrease in the gradient norm after pruning [59]. **sanity-check** proposes sanity check methods to identify whether the architecture of the pruned subnetwork are essential for the final performance [54]. **Continuous Sparsification** searches for sparse networks based on an approximation of an intractable $l_0$ regularization [53]. To our best knowledge, this work is the first to theoretically verify the LTH and the existence of winning tickets by leveraging dynamical systems theory and inertial manifold theory.

**Variants of IMC model.** We evaluate three variants of IMC to show the strengths of different subnetwork training strategies. After leveraging dynamical systems theory and inertial manifold theory to generate winning lottery tickets, IMC-Reinitialization randomly sample a new initialization for winning tickets' initialization [13], IMC-Rewinding rewinds unpruned weights to their values from earlier in training and retrains them from there using the original training schedule [15], and IMC-FineTuning fine-tunes the pruned subnetworks to regain the lost performance [38].

**Evaluation metrics.** We evaluate the proposed techniques against the baselines by focusing on two specific tasks: (1) the classification accuracy with the same sparsity level found by our IMC model; and (2) the pruning performance (i.e., sparsity) under the highest classification accuracy. A larger sparsity indicates a better pruning.

Table 3: Accuracy with found sparsity by our IMC method on ImageNet

| Neural Network | ResNet-20 | | | | ResNet-32 | | | |
|---|---|---|---|---|---|---|---|---|
| Metric | Sparsity | Accuracy | Epoch | Runtime (s) | Sparsity | Accuracy | Epoch | Runtime (s) |
| Baseline | 0 | 59.4 | 160 | 6hrs | 0 | 60.0 | 160 | 8hrs |
| Flow&Prune | 0.33 | 57.6 | 160 | 23,242 | 0.28 | 58.9 | 160 | 34,920 |
| SNIP | 0.33 | 55.5 | 160 | 18,100 | 0.28 | 56.6 | 160 | 28,947 |
| SynFlow | 0.33 | 56.6 | 160 | 17,990 | 0.28 | 56.1 | 160 | 25,145 |
| LTH+Reinitialization | 0.34 | 55.9 | 800 | 198,832 | 0.27 | 59.4 | 640 | 129,344 |
| LTH+Rewinding | 0.34 | 56.5 | 800 | 100,402 | 0.27 | 59.2 | 640 | 127,899 |
| LTH+FineTuning | 0.34 | 55.9 | 800 | 198,658 | 0.27 | 57.9 | 640 | 130,146 |
| GraSP | 0.33 | 55.4 | 160 | 20,374 | 0.28 | 56.0 | 160 | 29,287 |
| sanity-check | 0.33 | 57.4 | 160 | **10,573** | 0.28 | 58.8 | 160 | **16,288** |
| Continuous Sparsification | 0.3 | 54.1 | 320 | 21,285 | 0.3 | 55.6 | 320 | 18,336 |
| IMC-Reinitialization | 0.33 | **59.5** | 160 | 14,477 | 0.28 | **60.9** | 160 | 25,783 |
| IMC-Rewinding | 0.33 | 59.2 | 160 | 14,468 | 0.28 | 60.6 | 160 | 23,655 |
| IMC-FineTuning | 0.33 | 59.2 | 160 | 14,474 | 0.28 | 60.7 | 160 | 22,264 |

Table 4: Sparsity with highest accuracy on ImageNet

| Neural Network | ResNet-20 | | | | ResNet-32 | | | |
|---|---|---|---|---|---|---|---|---|
| Metric | Sparsity | Accuracy | Epoch | Runtime (s) | Sparsity | Accuracy | Epoch | Runtime (s) |
| Baseline | 0 | 59.4 | 160 | 6hrs | 0 | 60.0 | 160 | 8hrs |
| Flow&Prune | 0.1 | 58.9 | 160 | 23,242 | 0.1 | 59.3 | 160 | 34,920 |
| SNIP | 0.1 | 55.7 | 160 | 28,972 | 0.1 | 55.7 | 160 | 28,972 |
| SynFlow | 0.33 | 56.6 | 160 | 17,990 | 0.1 | 56.2 | 160 | 29,062 |
| LTH+Reinitialization | 0.1 | 59.1 | 320 | 79,479 | 0.1 | 60.4 | 320 | 66,160 |
| LTH+Rewinding | 0.1 | 58.7 | 320 | 40,219 | 0.1 | 60.2 | 320 | 65,455 |
| LTH+FineTuning | 0.1 | 58.0 | 320 | 79,661 | 0.1 | 60.5 | 320 | 65,979 |
| GraSP | 0.1 | 55.8 | 160 | 20,503 | 0.1 | 56.1 | 160 | 32,695 |
| sanity-check | 0.1 | 58.9 | 160 | 10,422 | 0.1 | 59.4 | 160 | 16,899 |
| Continuous Sparsification | 0.16 | 54.2 | 320 | 22,287 | 0.1 | 56.1 | 320 | 18,160 |
| IMC-Reinitialization | **0.33** | 59.5 | 160 | 14,477 | **0.28** | 60.9 | 160 | 25,783 |
| IMC-Rewinding | **0.33** | 59.2 | 160 | 14,468 | **0.28** | 60.6 | 160 | 23,655 |
| IMC-FineTuning | **0.33** | 59.2 | 160 | 14,474 | **0.28** | 60.7 | 160 | 22,264 |

**Accuracy of pruned subnetworks with found sparsity by our IMC method.** Tables 1 and 3 exhibit the accuracy of pruned subnetworks obtained by thirteenth network pruning and LTH approaches with found sparsity by our IMC method on two datasets. Baseline represents the accuracy achieved by the original unpruned neural networks. Since our IMC method executes a one-time pruning operation, the sparsity level generated by IMC is unique, e.g., 0.47 for ResNet-20 and 0.45 for ResNet-32 on CIFAR-10. We compare all methods with the same sparsity level. The LTH approaches prune $p\%$ of the parameters in each round and thus may not prune the networks with the exact same sparsity level found by our IMC method. In this case, we choose the closest sparsity level to prune the networks, say 0.47 for LTH+Rewinding with ResNet-32 on CIFAR-10. It is observed among thirteenth comparison methods, with the same sparsity level, three variants of our IMC achieve the highest accuracy in all experiments, showing the effectiveness of IMC to the neural network pruning. Compared to the accuracy by the Baseline method, IMC, on average, achieves only 0.1% accuracy loss on CIFAR-10, and even gains 0.3% accuracy improvement on ImageNet surprisingly.

**Sparsity of pruned subnetworks with highest accuracy.** Tables 2 and 4 present the sparsity of pruned subnetworks obtained by thirteenth methods with highest accuracy within a range of sparsity levels on two datasets. Compared to the sparsity by other methods, under highest accuracy by pruned subnetworks, IMC, on average, improves sparsity for 32.5% and 18.6% on CIFAR-10 and ImageNet respectively. In addition, even if three variants of our IMC method have the highest sparsity, the accuracy by them are still better than most comparison methods in most experiments. The promising performance of IMC over CIFAR-10 and ImageNet datasets implies that IMC has great potential as a general network pruning solution to other image datasets, which is desirable in practice.

**Ablation study.** Tables 1- 4 also compares the accuracy over two datasets with three variants of our IMC model with different initialization and training strategies of pruned subnetworks. We have observed that three versions of IMC achieves the similar accuracy (94.4-95.1) on CIFAR-10 and (59.2-60.7) over ImageNet. The difference of the accuracy achieved by three versions, on average, is only 2.5% on CIFAR-10 and 5.5% over ImageNet respectively. A reasonable explanation is that our IMC methods utilize the dynamical systems theory and inertial manifold theory to analyze the

Table 5: Accuracy with varying training epochs

| Neural Network | Accuracy | | | | |
|---|---|---|---|---|---|
| IMC-Reinitialization on CIFAR-10 | 94.27 | 94.51 | 94.57 | 94.18 | 94.37 |
| IMC-Rewinding on CIFAR-10 | 94.15 | 94.43 | 94.29 | 94.69 | 94.74 |
| IMC-FineTuning on CIFAR-10 | 94.52 | 94.56 | 94.54 | 94.96 | 94.51 |
| IMC-Reinitialization on ImageNet | 59.22 | 59.51 | 59.35 | 58.70 | 59.28 |
| IMC-Rewinding on ImageNet | 58.90 | 59.19 | 58.80 | 58.72 | 58.76 |
| IMC-FineTuning on ImageNet | 58.67 | 59.53 | 59.42 | 58.74 | 58.56 |
| IMC-Reinitialization on CIFAR-100 | 73.26 | 73.40 | 73.37 | 73.56 | 73.80 |
| IMC-Rewinding on CIFAR-100 | 73.46 | 73.35 | 73.18 | 73.54 | 73.53 |
| IMC-FineTuning on CIFAR-100 | 73.58 | 73.17 | 73.43 | 73.23 | 74.22 |

(a) CIFAR10  (b) ImageNet  (c) CIFAR100

Figure 1: Accuracy with varying sparsity levels

Figure 2: Accuracy with varying training epochs

dynamics and inertial manifold structure of our gradient system, instead of manually-crafted pruning heuristics. The above rigorous mathematical analysis substantially decreases the impact of different initialization and training strategies of pruned subnetworks.

**Running time.** Tables 1- 4 report the running time achieved by all comparison methods and three versions of our IMC over two dataset to produce pruned subnetworks respectively. Compared with pruning before training techniques, our IMC methods exchange the cost of reducing $W$ to $W^+$ with better sparsity and accuracy. Except pruning before training techniques, our IMC methods outperform all other types of network pruning and LTH techniques. Especially, our IMC methods show good efficiency for network pruning on ImageNet and only sanity-check presents better efficiency.

**Impact of sparsity levels.** Figure 1 measures the performance effect of sparsity levels for network pruning by varying relative sparsity from -20 to 20. Relative sparsity 0 corresponds to the absolute sparsity level found by our IMC, say $0.47$ on CIFAR-10. The absolute sparsity is equal to the sum of the relative sparsity and the sparsity level found by our IMC, e.g., the absolute sparsity is equal to $0.15 + 0.47$ for the relative sparsity $0.15$ on CIFAR-10. Although our IMC method executes a one-time pruning operation with a single found sparsity level, in order to verify whether the found sparsity level is the optimal, we measure the performance under different sparsity levels. The performance curves initially keeps stable when the sparsity increases. Later on, the performance curves decreases substantially when the sparsity continuously increases and is beyond the found sparsity level by our IMC. This validates the effectiveness of our inertial manifold-based network pruning for discovering the optimal sparsity level.

**Impact of training epochs.** Table 5 and Figure 2 evaluates the accuracy impact of training epochs on the original large networks. It is observed that when changing training epochs, the accuracy of pruned subnetworks by the IMC model keeps relatively stable, i.e., the accuracy fluctuates within the range of less than 1%. This shows the performance of pruned subnetworks is insensitive to the pre-training of the original networks. Especially, as shown in Table 5, limited training of the original networks is enough to help capture of the dynamics and inertial manifold structure of the gradient system, and thus produce a good network pruning.

# 5 Conclusions

In this work, we have theoretically verified the validity of the LTH. First, we explored the possibility of theoretically lossless pruning as well as one-time pruning. Second, we utilized the inertial manifold theory to reduce the original high-dimensional system to a low-dimensional system. Finally, we demonstrated the precondition and existence of pruned subnetworks and prune the original networks in terms of the gap in their spectrum that make the subnetworks have the smallest dimensions.

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
