# A  Supplementary Materials

## A.1  Related Work

**Neural Network Pruning.** A number of neural network pruning techniques have been proposed for reducing the computational expense of large-scale neural networks for deployment on low-resource systems. To estimate the impact of removing a parameter, these methods often use importance measures that were originally designed to prune neural networks. Existing techniques on neural network pruning can be broadly classified into three categories: (1) Pruning after training which consists of three steps: training the original network to convergence, prune redundant weights based on some criteria, and retrain the pruned model to regain the performance loss due to pruning [20, 62, 34, 23, 41, 67, 49, 68, 52, 57, 3, 30, 48, 60, 50, 36]; (2) Pruning during training techniques aim to update the pruning mask while training the network from scratch, which allows pruned neurons to be recovered [74, 24, 25, 63, 35, 64, 39, 27, 29, 4, 40]; and (3) Pruning before training approaches try to find the sparsity mask at initialization and train the pruned network from scratch without changing the mask [32, 61, 33, 56, 16, 21]. A recent literature has demonstrated that traditional neural network pruning often fall-short compared to the LTH techniques [33].

**Lottery Ticket Hypothesis.** The Lottery Ticket Hypothesis (LTH) states that a randomly-initialized, dense neural network contains a subnetwork that is initialized such that, when trained in isolation, it can match the test accuracy of the original network after training for at most the same number of iterations [13, 73, 70, 42, 6, 72, 37]. If this hypothesis is true, it has great potential to covert the inefficient training process on a large network to the scalable training process over a small one with comparable test accuracy. Most of existing LTH techniques provide empirical evidence to verify the LTH, although these methods raise very intriguing observations [71, 12, 1, 47, 69, 54, 5, 53, 26, 8, 7, 11]. Some LTH techniques utilize unstructured weight magnitude pruning to obtain the matching subnetworks [13, 70]. The original LTH papers utilize the Iterative Magnitude Pruning (IMP) algorithm to run multiple cycles of training, pruning, and weight rewinding to discover extremely sparse neural networks at initialization that can be trained to match the test accuracy of the original network [13, 14]. However, multiple cycles of training and pruning over large neural networks are time-consuming. In order to scale over large models and datasets, a late-rewinding technique is proposed to reinitialize ticket training from the early training stage rather than rewinding to initialization [14, 49, 15]. Other methods employ the loss gradients at initialization to prune the network in a single-shot for improving the LTH efficiency [32, 59]. Two recent works analyze the LTH transferability, i.e., the ticket discovered from one source task can be transferred to another target task [44, 43].

Several recent works attempt to theoretically verify the LTH [42, 45, 46, 48]. A common characteristic of these methods is to validate that target neural networks can be approximated as pruned subnetworks with bounded error with a certain success probability (i.e., confidence). This implies that they only discuss the validity of approximate subnetworks (i.e., lossy pruning) and may fail to generate a good approximation with some failure probability. In addition, these findings cannot offer much insights for how to develop effective pruning algorithms that are applied on the target network directly. The proof strategies rely on the heuristics that a subnetwork with a specific structure that can replicate a single neuron of the target network exists.

To our best knowledge, this work is the first to theoretically verify the LTH and the existence of winning lottery tickets by leveraging dynamical systems theory and inertial manifold theory. We theoretically identify the precondition and existence of exact pruned subnetworks (i.e., one-time lossless pruning) with 100% confidence and directly prune the original neural networks in terms of the gap in their spectrum.

## A.2  Proof of Theorems

**Lemma 1.** *If $f(x)$ is globally Lipschitz in $X$, then the ODE in Eq.(1) has a unique global solution in $X$ for any initial condition $x(0) = x_0$.*

*Proof. Please refer to the book [9] for detailed proof.*

**Theorem 1.** *The global attractor $A$ of the dynamical system derived by Eq.(7) is the union of equilibria $W^*$ and their unstable manifolds $\mathcal{M}^u(W^*)$ [19].*

*Proof. Let $\varphi(t, W(0))$ be the dynamical system derived by Eq.(7). For any initial value $W(0) \in \mathbb{R}^d$, the $\Omega$-limit set of $W(0)$ is defined as follows.*

$$\Omega(W(0)) = \{\bar{W} \in \mathbb{R}^d : \exists t_n \to \infty \text{ as } n \to \infty, \text{ s.t. } \varphi(t_n, W(0)) \to \bar{W}\} \tag{24}$$

*For any $\bar{W} \in \Omega(W(0))$ and any $t \geq 0$, when $n \to \infty$, we have*

$$\varphi(t + t_n, W(0)) = \varphi(t, \varphi(t_n, W(0))) \to \varphi(t, \bar{W}) \tag{25}$$

*This implies $\varphi(t, \bar{W}) \in \Omega(W(0))$ and further indicates that $\Omega(W(0))$ is invariant under the dynamical system $\varphi$. Since $L(\varphi(t, W(0)))$ that decreases with $t$ and the loss function $L$ has a lower bound, there exists a constant $D$ such that*

$$L(\phi(t, W(0))) \to D \text{ as } t \to \infty \tag{26}$$

*According to the definition of $\Omega(W(0))$, we have $L(\bar{W}) = D$ for any $\bar{W} \in \Omega(W(0))$. Since $\varphi(t, \bar{W}) \in \Omega(W(0))$, for any $t \geq 0$, we get*

$$L(\varphi(t, \bar{W})) = L(\bar{W}) \tag{27}$$

*This implies*

$$\frac{d}{dt} L(\varphi(t, \bar{W})) = -|\nabla L(\phi(t, \bar{W}))|^2 = 0 \tag{28}$$

*Therefore, for any $\bar{W} \in \Omega(W(0))$, $\nabla L(\bar{W}) = 0$, i.e., $\bar{W}$ is an equilibrium of $\varphi$. This indicates that for any $W \in \mathbb{R}^d$, $\varphi(t, W) \to W^*$ for a certain equilibrium $W^*$. Since the unstable manifold $\mathcal{M}^u(W^*)$ of $W^*$ is attracting all orbits in a neighborhood of $W^*$, the global attractor $A$ contains all equilibria $W^*$ and their unstable manifolds $\mathcal{M}^u(W^*)$.*

**Theorem 2.** *The global attractor $A$ of the dynamical system derived by Eq.(7) is unique.*

*Proof. Let $A_1$ and $A_2$ be two global attractors. Since $A_2$ is bounded, it is attracted by $A_1$, we have*

$$\text{dist}\left(\varphi(t, A_2), A_1\right) \to 0 \text{ as } t \to \infty \tag{29}$$

*In addition, $A_2$ is invariant, $\varphi(t, A_2) = A_2$, from which it follows that $\text{dist}\left(A_2, A_1\right) = 0$. Notice that the statement is symmetric, so $\text{dist}\left(A_1, A_2\right) = 0$, from which it follows that $A_1 = A_2$.*

**Theorem 3.** *Let $A \subset \mathcal{W}$ be a compact subset of a Banach space $\mathcal{W}$, and $W(t) \in \mathcal{W}$ be a sequence with $\lim_{t \to \infty} \text{dist}(W(t), A) = 0$, then $W(t)$ has a convergent subsequence, whose limit lies in $A$.*

*Proof. Write $W(t) = R(t) + S(t)$, where $R(t) \in A$ and $|S(t)| \to 0$ as $t \to \infty$. Then there is a subsequence such that $R(t_j) \to R^* \in A$, so $W(t_j) \to R^*$ too.*

**Theorem 4.** *The global attractor $A$ of the dynamical system derived by Eq.(7) is the maximal compact invariant set, and the minimal set that attracts all bounded sets $B \subset \mathcal{W}$.*

*Proof. Let $X$ be compact and invariant. Since $X$ is compact, it is bounded and thus is attracted to $A$. Therefore, $\text{dist}(\varphi(t, X), A) = \text{dist}(X, A) \to 0$ as $t \to \infty$, i.e., $\text{dist}(X, A) = 0$ so $X \subseteq A$. Similarly, if $Y$ attracts all bounded sets, then $Y$ attracts $A$. The same argument shows that $A \subseteq Y$ by using invariance of $A$. In conclusion, $X \subseteq A \subseteq Y$.*

**Theorem 5.** *Let $\mathcal{W}$ be a Banach space with dimension $d$, given a semi-linear ODE*

$$W'(t) = AW(t) + f(W(t)), \ W(t) \in \mathcal{W} \tag{30}$$

*where $A$ is a $d \times d$ matrix and $f : \mathcal{W} \to \mathcal{W}$. The spectrum of $A$ is denoted as follows.*

$$\lambda_1 \leq \lambda_2 \leq \cdots \leq \lambda_d \tag{31}$$

*Assuming that $f(0) = 0$, let $K$ be the Lipschitz constant of $f(x)$, that is,*

$$\|f(x) - f(y)\|_\mathcal{W} \leq K\|x - y\|_\mathcal{W} \text{ for any } x, y \in \mathcal{W} \tag{32}$$

*If there exits an integer $k$, such that $\lambda_{k+1} - \lambda_k > 4K$, let $\mathcal{W}^+$ be the generalized eigenspace for $\lambda_{k+1}, \ldots, \lambda_d$ and $\mathcal{W}^-$ be the generalized eigenspace for $\lambda_1, \ldots, \lambda_k$, then there exists an inertial manifold $\mathcal{M}_\mathcal{W}$ as follows.*

$$\mathcal{M}_\mathcal{W} = (\mathcal{W}^+, h(\mathcal{W}^+)) \tag{33}$$

*where $h$ is a mapping from $\mathcal{W}^+$ to $\mathcal{W}^-$.*

*Proof. Let $\Pi^+$ and $\Pi^-$ be the projection maps associated with subspaces $\mathcal{W}^+$ and $\mathcal{W}^-$ respectively. $\Pi^+$ and $\Pi^-$ satisfy*

$$\Pi^+\mathcal{W} = \mathcal{W}^+, \ \Pi^-\mathcal{W} = \mathcal{W}^-, \ \Pi^+\Pi^+ = \Pi^+, \ \Pi^-\Pi^- = \Pi^- \tag{34}$$

*Now, we view the matrix $A$ as a linear operator defined with matrix multiplication. In order to study the ODE in $\mathcal{W}^+$ and $\mathcal{W}^-$ respectively, let*

$$A^+ = A|_{\mathcal{W}^+} = A\Pi^+, \ A^- = A|_{\mathcal{W}^-} = A\Pi^- \tag{35}$$

*and*

$$f^+(W(t)) = \Pi^+ f(W(t)), \ f^-(W(t)) = \Pi^- f(W(t)) \tag{36}$$

*By using the spectrum gap condition, we have the following exponential dichotomy.*

$$\begin{aligned}
\|e^{A^- t}\|_{op} &\leq Ce^{\lambda_k t} \text{ for any } t \geq 0 \\
\|e^{A^+ t}\|_{op} &\leq Ce^{\lambda_{k+1} t} \text{ for any } t \leq 0
\end{aligned} \tag{37}$$

*where $\| \cdot \|_{op}$ is the operator norm from $\mathcal{W}$ to $\mathcal{W}$.*

*For any $W(t) \in \mathcal{W}$, we decompose $W(t)$ into two subspaces.*

$$W(t) = W(t)^+ + W(t)^- \tag{38}$$

*where $W(t)^+ = \Pi^+ W(t)$ and $W(t)^- = \Pi^- W(t)$.*

*Based on the decomposition of $W(t)$, we rewrite $f^+(W(t))$ and $f^-(W(t))$ as $f^+(W^+(t), W^-(t))$ and $f^-(W^+(t), W^-(t))$ respectively. Thus, by applying $\Pi^+, \Pi^-$ to Eq.(30), the semi-linear ODE is rewritten as follows.*

$$\begin{cases} \frac{d}{dt}W^+(t) = A^+W^+(t) + f^+(W^+(t), W^-(t)) \\ \frac{d}{dt}W^-(t) = A^-W^-(t) + f^-(W^+(t), W^-(t)), \end{cases} \tag{39}$$

*By utilizing the Constant Variation formula, we rewrite Eq.(39) in the integral form .*

$$W^+(t) = e^{A^+ t} W^+(0) + \int_0^t e^{A^+(t-s)} f^+(W^+(s), W^-(s)) ds$$

$$W^-(t) = e^{A^-(t-t_0)} W^-(t_0) + \int_{t_0}^t e^{A^-(t-s)} f^-(W^+(s), W^-(s)) ds$$

(40)

*Now, we employ the Lyapunov-Perron method to construct the inertial manifold. Let $\alpha = (\lambda_k + \lambda_{k+1})/2$, a weighted norm $\| \cdot \|_\alpha$ on continuous function space $C(\mathbb{R}^-; \mathcal{W})$ is defined as follows.*

$$\|P(t)\|_\alpha = \sup_{t \leq 0} e^{-\alpha t} \|P(t)\|_\mathcal{W}, \ P(t) \in C(\mathbb{R}^-; \mathcal{W})$$

(41)

*We define $\mathcal{W}_\alpha$ as a continuous function space $C(\mathbb{R}^-; \mathcal{W})$ equipped with $\| \cdot \|_\alpha$ norm.*

$$\mathcal{W}_\alpha(\delta) = \{W(t) \in \mathcal{W}_\alpha | \|P\|_\alpha \leq \delta, \|W^+(0)\|_\mathcal{W} \leq \delta/2\}$$

(42)

*For any $(W^+(t), W^-(t)) \in \mathcal{W}_\alpha(\delta)$, we define*

$$\tilde{W}^+(t) = e^{A^+ t} W^+(0) + \int_0^t e^{A^+(t-s)} f^+(W^+(s), W^-(s)) ds$$

(43)

$$\tilde{W}^-(t) = e^{A^-(t-t_0)} W^-(t_0) + \int_{t_0}^t e^{A^-(t-s)} f^-(W^+(s), W^-(s)) ds$$

(44)

*By applying the dichotomy property in Eq.(37) to the linear term in Eq.(44), we obtain*

$$\|e^{A^-(t-t_0)} W^-(t_0)\|_\mathcal{W} \leq C e^{\lambda_k(t-t_0)} \|W^-(t_0)\|_\mathcal{W} \text{ for } t_0 \leq t \leq 0$$

(45)

*By multiplying the weight $e^{-\alpha t}$ to Eq.(45), we have*

$$e^{-\alpha t} \|e^{A^-(t-t_0)} W^-(t_0)\|_\mathcal{W} \leq C e^{(\lambda_k - \alpha)(t-t_0)} \left( e^{-\alpha t_0} \|W^-(t_0)\|_\mathcal{W} \right)$$
$$\leq C e^{(\lambda_k - \alpha)(t-t_0)} \|W^-(t)\|_\alpha$$

(46)

*This implies*

$$\|e^{A^-(t-t_0)} W^-(t_0)\|_\alpha \leq C e^{(\lambda_k - \alpha)(t-t_0)} \|W^-(t)\|_\alpha$$

(47)

*By taking $t_0 \to -\infty$, we have*

$$\|e^{A^-(t-t_0)} W^-(t_0)\|_\alpha \to 0$$

(48)

*Therefore, letting $t_0 \to -\infty$ in Eq.(44), we have*

$$\tilde{W}^-(t) = \int_{-\infty}^t e^{A^-(t-s)} f^-(W^+(s), W^-(s)) ds$$

(49)

*Now, we consider the system consisting of Eqs.(43) and (49), i.e.,*

$$\begin{cases} \tilde{W}^+(t) &= e^{A^+ t} W^+(0) + \int_0^t e^{A^+(t-s)} f^+(W^+(s), W^-(s)) ds \\ \tilde{W}^-(t) &= \int_{-\infty}^t e^{A^-(t-s)} f^-(W^+(s), W^-(s)) ds \end{cases}$$

(50)

*The above system can be viewed as a mapping $\Gamma : (W^+(t), W^-(t)) \to (\tilde{W}^+(t), \tilde{W}^-(t))$. We will use the Banach fixed-point theorem to prove there exists a unique fixed point for the mapping $\Gamma$.*

*By using the dichotomy property in Eq.(37) and the properties of $f$ in Eq.(43), we get that for any $t \leq 0$*

$$
\begin{aligned}
\|\tilde{W}^+(t)\|_{\mathcal{W}} \leq & e^{\lambda_{k+1}t}\|W^+(0)\|_{\mathcal{W}} + \int_0^t e^{\lambda_{k+1}(t-s)}\|f^+(W^+(s), W^-(s))\|_{\mathcal{W}} ds \\
\leq & e^{\lambda_{k+1}t}\|W^+(0)\|_{\mathcal{W}} + K^+ \int_0^t e^{\lambda_{k+1}(t-s)}(\|W^+(s)\|_{\mathcal{W}} + \|W^-(s))\|_{\mathcal{W}}) ds
\end{aligned}
\tag{51}
$$

*where $K^+$ be the Lipschitz constant of $f^+(x)$.*

*By multiplying $e^{-\alpha t}$ to Eq.(51), we obtain*

$$
\begin{aligned}
\|\tilde{W}^+(t)\|_\alpha \leq & e^{(\lambda_{k+1}-\alpha)t}\|W^+(0)\|_{\mathcal{W}} + K^+ \int_0^t e^{(\lambda_{k+1}-\alpha)(t-s)} e^{-\alpha s}(\|W^+(s)\|_{\mathcal{W}} + \|W^-(s))\|_{\mathcal{W}}) ds \\
\leq & \|W^+(0)\|_{\mathcal{W}} + K^+(\|W^+(t)\|_\alpha + \|W^-(t))\|_\alpha) \int_{-\infty}^0 e^{(\lambda_{k+1}-\alpha)(t-s)} ds \\
\leq & \delta/2 + K^+(\lambda_{k+1}-\alpha)^{-1}(\|W^+(t)\|_\alpha + \|W^-(t)\|_\alpha)
\end{aligned}
\tag{52}
$$

*where we use $\|W^+(0)\|_\alpha \leq \delta/2$ as the definition of $\mathcal{W}_\alpha(\delta)$ in Eq.(42).*

*Similarly, for Eq.(49), we have*

$$
\|\tilde{W}^-(t)\|_\alpha \leq K^-(\alpha - \lambda_k)^{-1}(\|W^+(t)\|_\alpha + \|W^-(t)\|_\alpha)
\tag{53}
$$

*where $K^-$ be the Lipschitz constant of $f^-(x)$.*

*Since $\alpha = (\lambda_k + \lambda_{k+1})/2$, we have $\alpha - \lambda_k > 2K$ and $\lambda_{k+1} - \alpha > 2K$. Therefore, $\lambda_{k+1} - \lambda_k > 4K$. In addition, as $f = f^+ + f^-$, we have $K^+ + K^- = K$. Therefore, by combining Eqs.(52) and (53), we obtain*

$$
\begin{aligned}
\|\tilde{W}^+(t)\|_\alpha + \|\tilde{W}^-(t)\|_\alpha < & \delta/2 + \frac{(K^+ + K^-)}{2K}(\|W^+(t)\|_\alpha + \|W^-(t)\|_\alpha) \\
\leq & \delta/2 + \frac{(K^+ + K^-)}{2K}(\delta/2 + \delta/2) \\
= & \delta
\end{aligned}
\tag{54}
$$

*This implies the mapping $\Gamma$ maps $\mathcal{W}_\alpha(\delta)$ to itself. By taking $(W_i^+(t), W_i^-(t)) \in \mathcal{W}_\alpha(\delta)$ for $i = 1, 2$ that satisfies $W_1^+(0) = W_2^+(0)$, $(\tilde{W}_i^+(t), \tilde{W}_i^-(t))$ has the same definition as the one in Eq.(50), i.e.,*

$$
\begin{cases}
\tilde{W}_i^+(t) & = e^{A^+t}W_i^+(0) + \int_0^t e^{A^+(t-s)} f^+(W_i^+(s), W_i^-(s)) ds \\
\tilde{W}_i^-(t) & = \int_{-\infty}^t e^{A^-(t-s)} f^-(W_i^+(s), W_i^-(s)) ds
\end{cases}
\tag{55}
$$

*By following the similar strategy in Eqs.(52) and (53), we have*

$$
\|\tilde{W}_1^+(t) - \tilde{W}_2^+(t)\|_\alpha \leq K^+(\lambda_{k+1}-\alpha)^{-1}(\|W_1^+(t) - W_2^+(t)\|_\alpha + \|W_1^-(t) - W_2^-(t)\|_\alpha)
\tag{56}
$$

*and*

$$\|\tilde{W}_1^-(t) - \tilde{W}_2^-(t)\|_\alpha \leq K^-(\alpha - \lambda_k)^{-1}(\|W_1^+(t) - W_2^+(t)\|_\alpha + \|W_1^-(t) - W_2^-(t)\|_\alpha) \quad (57)$$

*By combining Eqs.(56) and (57) together, we get*

$$
\begin{aligned}
&\|\tilde{W}_1^+(t) - \tilde{W}_2^+(t)\|_\alpha + \|\tilde{W}_1^-(t) - \tilde{W}_2^-(t)\|_\alpha \\
&\leq (K^+(\lambda_{k+1} - \alpha)^{-1} + K^-(\alpha - \lambda_k)^{-1})(\|W_1^+(t) - W_2^+(t)\|_\alpha + \|W_1^-(t) - W_2^-(t)\|_\alpha) \\
&< \frac{\|W_1^+(t) - W_2^+(t)\|_\alpha + \|W_1^-(t) - W_2^-(t)\|_\alpha}{2}
\end{aligned}
\quad (58)
$$

*This implies that $\Gamma$ is a contraction mapping. Therefore, for any $W^+(0)$, $\Gamma$ admits a unique fixed point $(W^{*+}(t), W^{*-}(t))$, i.e.,*

$$
\begin{cases}
W^{*+}(t) = e^{A^+ t}W^+(0) + \int_0^t e^{A^+(t-s)} f^+(W^{*+}(s), W^{*-}(s))ds \\
W^{*-}(t) = \int_{-\infty}^t e^{A^-(t-s)} f^-(W^{*+}(s), W^{*-}(s))ds
\end{cases}
\quad (59)
$$

*We define a mapping $h$ from $\mathcal{W}^+$ to $\mathcal{W}^-$ as follows.*

$$h(W^+) = W^{*-}(0) \quad (60)$$

*where $(W^{*+}(t), W^{*-}(t))$ is the fixed point of $\Gamma$ (i.e., the solution of Eq.(59)) with $W^+(0) = W^+$. Then the inertial manifold $\mathcal{M}_\mathcal{W}$ is given as follows.*

$$\mathcal{M}_\mathcal{W} = \{(W^+, h(W^+)) \mid W^+ \in \mathcal{W}^+\} = (\mathcal{W}^+, h(\mathcal{W}^+)) \quad (61)$$

*Therefore, the proof is concluded.*

### A.3 Additional Experiments

**Accuracy and sparsity of pruned subnetworks over CIFAR-100.** Table 6 exhibits the accuracy of pruned subnetworks obtained by thirteenth network pruning and LTH approaches with found sparsity by our IMC method on CIFAR-100. Table 7 presents the sparsity of pruned subnetworks obtained by thirteenth methods with highest accuracy within a range of sparsity levels over CIFAR-100. Similar trends are observed for the neural network pruning comparison in these two tables: our IMC method achieves the largest accuracy values (>73.9) with the same sparsity levels and the largest sparsity (>0.18) with the highest accuracy, which are better than other nine baseline methods in all tests. Notice that even if the noise level is very high, such as 0.29, three versions of our IMC method still can achieve considerable accuracy improvement. It demonstrates that IMC is relatively robust to sparsity levels. This advantage is very important for the usage of deep learning models in resource-intensive scenarios with the requirement of low latency and energy consumption, such as Internet of Things and mobile computing.

Table 6: Accuracy with found sparsity by our IMC method on CIFAR-100

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

Table 12: Accuracy with found sparsity by our IMC method on CIFAR-100 (reduced training samples)

| Neural Network | ResNet-20 | | | | ResNet-32 | | | |
|---|---|---|---|---|---|---|---|---|
| Metric | Sparsity | Accuracy | Epoch | Runtime (s) | Sparsity | Accuracy | Epoch | Runtime (s) |
| Baseline | 0 | 40.7 | 160 | 539 | 0 | 41.3 | 160 | 685 |
| Flow&Prune | 0.13 | 33.7 | 160 | 1,293 | 0.22 | 30.5 | 160 | 1,589 |
| SNIP | 0.13 | 31.5 | 160 | **508** | 0.22 | 30.0 | 160 | 722 |
| SynFlow | 0.13 | 31.4 | 160 | 510 | 0.22 | 28.2 | 160 | 697 |
| LTH+Reinitialization | 0.10 | 38.1 | 320 | 1,083 | 0.19 | 38.1 | 480 | 2,263 |
| LTH+Rewinding | 0.10 | 37.5 | 320 | 1,100 | 0.19 | 38.7 | 480 | 2,362 |
| LTH+FineTuning | 0.10 | 39.2 | 320 | 1,086 | 0.19 | 38.1 | 480 | 2,377 |
| GraSP | 0.13 | 25.1 | 160 | 891 | 0.22 | 19.8 | 160 | 1,315 |
| sanity-check | 0.13 | 36.0 | 160 | 534 | 0.22 | 34.3 | 160 | **688** |
| Continuous Sparsification | 0.13 | 36.0 | 320 | 1,071 | 0.22 | 32.8 | 320 | 1,344 |
| IMC-Reinitialization | 0.13 | 40.6 | 160 | 587 | 0.22 | 40.7 | 160 | 718 |
| IMC-Rewinding | 0.13 | 40.3 | 160 | 578 | 0.22 | 41.7 | 160 | 722 |
| IMC-FineTuning | 0.13 | **40.9** | 160 | 557 | 0.22 | **42.4** | 160 | 704 |

Table 13: Sparsity with highest accuracy on CIFAR-100 (reduced training samples)

| Neural Network | ResNet-20 | | | | ResNet-32 | | | |
|---|---|---|---|---|---|---|---|---|
| Metric | Sparsity | Accuracy | Epoch | Runtime (s) | Sparsity | Accuracy | Epoch | Runtime (s) |
| Baseline | 0 | 40.7 | 160 | 539 | 0 | 41.3 | 160 | 685 |
| Flow&Prune | 0.10 | 33.7 | 160 | 1,305 | 0.10 | 31.7 | 160 | 1,523 |
| SNIP | 0.10 | 31.7 | 160 | 505 | **0.22** | 30.0 | 160 | 728 |
| SynFlow | 0.10 | 31.4 | 160 | 510 | 0.10 | 30.0 | 160 | 704 |
| LTH+Reinitialization | 0.10 | 38.1 | 320 | 1,083 | 0.10 | 38.1 | 480 | 2,341 |
| LTH+Rewinding | 0.10 | 37.5 | 320 | 1,100 | 0.19 | 38.7 | 480 | 2,362 |
| LTH+FineTuning | 0.10 | 39.2 | 320 | 1,086 | 0.19 | 38.1 | 320 | 1,575 |
| GraSP | 0.10 | 26.7 | 160 | 872 | 0.10 | 26.6 | 160 | 1,368 |
| sanity-check | 0.13 | 36.0 | 160 | 534 | 0.10 | 35.3 | 160 | 688 |
| Continuous Sparsification | 0.13 | 36.0 | 320 | 1,071 | 0.08 | 33.9 | 320 | 1,344 |
| IMC-Reinitialization | **0.13** | 40.6 | 160 | 587 | **0.22** | 40.7 | 160 | 718 |
| IMC-Rewinding | **0.13** | 40.3 | 160 | 578 | **0.22** | 41.7 | 160 | 722 |
| IMC-FineTuning | **0.13** | 40.9 | 160 | 557 | **0.22** | 42.4 | 160 | 704 |

**Accuracy and sparsity of pruned subnetworks over datasets with reduced training samples.** In order to perform a comprehensive study about the applicability of neural network pruning by our IMC method under limited training samples. We randomly decrease the number of training samples over CIFAR-10, ImageNet, and CIFAR-100 by 90%, i.e., reduce the number of training samples over CIFAR-10, ImageNet, and CIFAR-100 from 50,000, 100,000, and 50,000 to 5,000, 10,000, and 5,000 respectively. The number of test samples keep unchanged, 10,000, 10,000, and 10,000 on CIFAR-10, ImageNet, and CIFAR-100 respectively. Tables 8-13 shows the accuracy and sparsity of pruned subnetworks obtained by thirteenth network pruning and LTH approaches over three datasets respectively. We have observed similar results, i.e., the generated pruned networks by three variants of our IMC method achieve the best accuracy and sparsity in most experiments, showing the superior performance of IMC in the presence of limited training samples. A reasonable explanation is that the rigorous mathematical analysis based on the dynamical systems theory and inertial manifold theory substantially improves the effectiveness and applicability of our IMC method in different scenarios.

## A.4  Parameter Sensitivity

In this section, we conduct more experiments to validate the sensitivity of various parameters in our IMC method for the neural network pruning task.

**Impact of rewinding strategies.** IMC-Rewinding rewinds unpruned weights to their values from earlier in training and retrains them from there using the original training schedule. Figure 3 measures the performance effect of the $i^{th}$ training epochs of the original large neural networks in which IMC-Rewinding rewinds the unpruned parameters to the ones in the $i^{th}$ training epochs of the original large neural networks. We have observed that the accuracy curves keep relatively stable when we

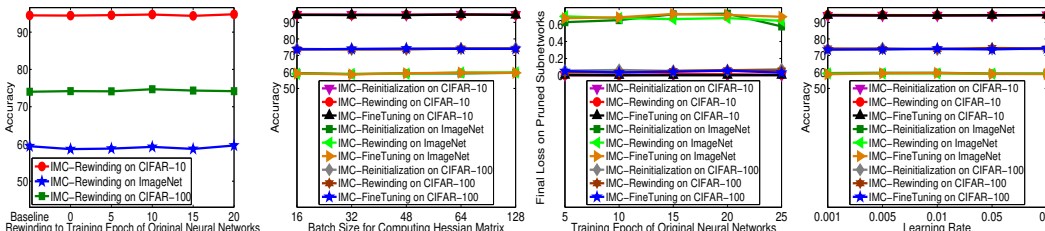

Figure 3: Accuracy Figure 4: Accuracy with Figure 5: Final loss on Figure 6: Accuracy with with varying rewinding varying batch sizes for subnetworks with vary- varying learning rates of epochs Hessian matrix ing training epochs neural networks

continuously change the rewinded training epochs. This demonstrates that our IMC-Rewinding method is insensitive to the parameter initialization of pruned subnetworks. No matter what the rewinded training epochs are, our IMC-Rewinding method can always achieve the superior accuracy in all tests, showing the effectiveness of IMC-Rewinding to the neural network pruning.

**Sensitivity of batch sizes for Hessian matrix.** Theorem 5 demonstrates the whole solution space $\mathcal{W}$ can be decomposed into two subspaces $\mathcal{W}^+$ and $\mathcal{W}^-$ when the gap between arbitrary two consecutive eigenvalues in the spectrum of $H(W^*)$ is larger than $4K$. Due to large size of $W^*$, we employ a scalable Hessian computation method, PyHessian, to calculate an approximate $H(W^*)$ [65, 66]. Figure 4 exhibits the impact of different batch sizes in training PyHessian by varying batch size between 16 and 128. The x-axis shows different values of batch sizes. It is observed that the accuracy values are stable with varying batch sizes. PyHessian computes approximate Hessian matrix based on the rigorous theory of numerical linear algebra (NLA) and randomized NLA, making it less sensitive to the batch size. In addition, PyHessian supports distributed implementation-allowing distributed-memory execution on both cloud and supercomputer systems, for fast and efficient Hessian computation. This significantly improves the applicability of our neural network pruning method.

**Convergence study.** Figure 5 presents the convergence of retraining the pruned subnetworks after different training epochs of the original large neural networks. As we can see, the accuracy scores keep stable when we change the number of the training epochs of the original large neural networks. The final loss by three versions of our IMC model can always converge to the same points with different training epochs of the original large neural networks. A rational guess that the rigorous mathematical analysis based on the dynamical systems theory and inertial manifold theory can always help our IMC model achieve the convergence on three datasets. This verifies the effectiveness of the IMC method for neural network pruning.

**Influence of learning rates.** Figure 6 shows the influence of learning rate in our IMC model by varying it from 0.001 to 0.1. It is observed that the accuracy values are stable with varying learning rate. Namely, our dynamical systems theory and inertial manifold theory-based neural network pruning method is insensitive to learning rate in the training of both the original neural networks and the pruned subnetworks. This demonstrates that our IMC model can always result in the good classification accuracy for neural network pruning while maintaining good efficiency, no matter which learning rate is selected.

**Impact of parameters $p$ and $\Delta$.** Tables 14 and 15 exhibit the impact of different $p$ with fixed $\Delta$ and the influence of different $\Delta$ with fixed $p$ in the estimation of Lipschitz constant of $F(U)$ by utilizing Eq.(23). In the current experiments, the estimated Lipschitz constants of all neural networks are very tiny with order $< 10^{-12}$. This implies that most of neural networks are easy to meet the gap condition about its spectrum and Lipschitz constant in Theorem 5. Therefore, our neural network pruning technique has great potential as a general pruning solution to other neural networks, which is desirable in practice.

Table 14: Estimated Lipschitz constant of $F(U)$ with PyHessian ($\Delta = 0.01$)

| $p$ | 2 | 5 | 10 |
|---|---|---|---|
| CIFAR-10 | $6.17 \times 10^{-12}$ | $< 1 \times 10^{-38}$ | $< 1 \times 10^{-38}$ |
| ImageNet | $1.14 \times 10^{-24}$ | $< 1 \times 10^{-38}$ | $< 1 \times 10^{-38}$ |
| CIFAR-100 | $2.57 \times 10^{-22}$ | $< 1 \times 10^{-38}$ | $< 1 \times 10^{-38}$ |

Table 15: Estimated Lipschitz constant of $F(U)$ with PyHessian ($p = 2$)

| $\Delta$ | 0.01 | 0.05 | 0.1 | 0.5 | 1 |
|---|---|---|---|---|---|
| CIFAR-100 | $6.17 \times 10^{-12}$ | $1.95 \times 10^{-17}$ | $3.72 \times 10^{-19}$ | $1.39 \times 10^{-22}$ | $5.93 \times 10^{-24}$ |
| ImageNet | $1.14 \times 10^{-24}$ | $3.25 \times 10^{-28}$ | $1.08 \times 10^{-29}$ | $3.69 \times 10^{-33}$ | $1.15 \times 10^{-34}$ |
| CIFAR-100 | $2.57 \times 10^{-22}$ | $8.17 \times 10^{-26}$ | $4.14 \times 10^{-27}$ | $1.32 \times 10^{-30}$ | $2.69 \times 10^{-32}$ |

## A.5 Experimental Details

**Environment.** The experiments were conducted on a compute server running on Red Hat Enterprise Linux 7.2 with 2 CPUs of Intel Xeon E5-2650 v4 (at 2.66 GHz) and 8 GPUs of NVIDIA GeForce GTX 2080 Ti (with 11GB of GDDR6 on a 352-bit memory bus and memory bandwidth in the neighborhood of 620GB/s), 256GB of RAM, and 1TB of HDD. Overall, the experiments took about 10 days in a shared resource setting. We expect that a consumer-grade single-GPU machine (e.g., with a 2080 Ti GPU) could complete the full set of experiments in around 17-18 days, if its full resources were dedicated.

**Training.** We study image classification networks on three standard image datasets: CIFAR-10 [3], CIFAR-100 [4], and ImageNet [5]. The above three image datasets are all public datasets, which allow researchers to use for non-commercial research and educational purposes. We train the baseline classifiers on the CIFAR-10/100 training set and test it on the CIFAR-10/100 test set. We use a subsample of 100,000 examples as training data and 10,000 examples as test data for ImageNet. We apply the ResNet-20 and ResNet-32 architectures for the CIFAR-10/100 and ImageNet datasets respectively [6]. The neural networks are trained with Kaiming initialization [22] using SGD for 160 epochs with an initial learning rate of 0.1 and batch size 100. The learning rate is decayed by a factor of 0.1 at 1/2 and 3/4 of the total number of epochs. In addition, we run each experiment for 3 trials for obtaining more stable results.

**Implementation.** For three neural network pruning models of Flow&Prune [7], SNIP [8], and SynFlow [9], we used the open-source implementation and default parameter settings by the original authors for the experiments. All models were run for 160 epochs, with a batch size of 100, and a learning rate of 0.1. For six state-of-the-art LTH approaches of LTH+Reinitialization [10], LTH+Rewinding [11], LTH+FineTuning [12], GraSP [13], sanity-check [14], and Continuous Sparsification [15], we also utilized the same model architecture as the official implementation provided by the original authors for neural network pruning in all experiments. All hyperparameters are standard values from reference codes or prior works. The above open-source codes from the GitHub are licensed under the MIT License, which only requires preservation of copyright and license notices and includes the permissions of commercial use, modification, distribution, and private use.

For our IMC model, we performed hyperparameter selection by performing a parameter sweep on parameter $p \in \{1, 2, 5, 10, \infty\}$ in the estimation of Lipschitz constant, $\Delta \in \{0.0001, 0.001, 0.01, 0.1\}$ in the estimation of Lipschitz constant, training epochs of the original large neural networks $\in \{5, 10, 20, 25, 30\}$ in LTH+Reinitialization and LTH+FineTuning, rewinding parameters to the ones in the training epochs of the original large neural networks $\in \{0.01, 0.05, 0.1, 0.2, 0.5\}$ in LTH+Rewinding, batch size for computing the approximate Hessian Matrix $\in \{16, 32, 48, 64, 128\}$, batch size for training the neural networks $\in \{20, 50, 100, 150, 200, 250\}$, and learning rate

---

[3] https://www.cs.toronto.edu/~kriz/cifar.html

[4] https://www.cs.toronto.edu/~kriz/cifar.html

[5] https://www.image-net.org/download.php

[6] https://github.com/KaimingHe/deep-residual-networks

[7] https://github.com/EkdeepSLubana/flowandprune

[8] https://github.com/namhoonlee/snip-public

[9] https://github.com/ganguli-lab/Synaptic-Flow

[10] https://github.com/rahulvigneswaran/Lottery-Ticket-Hypothesis-in-Pytorch

[11] https://github.com/facebookresearch/open_lth

[12] https://github.com/Eric-mingjie/rethinking-network-pruning

[13] https://github.com/alecwangcq/GraSP

[14] https://github.com/JingtongSu/sanity-checking-pruning

[15] https://github.com/lolemacs/continuous-sparsification

$\in \{0.0001, 0.0005, 0.001, 0.005, 0.01, 0.05, 0.1\}$. We select the best parameters over 50 epochs of training and evaluate the model at test time.

Table 16: Hyperparameter Settings

| Parameter | Value |
|---|---|
| Parameter $p$ in the estimation of Lipschitz constant | 2 |
| Parameter $\Delta$ in the estimation of Lipschitz constant | 0.01 |
| Training data ratio on CIFAR-10/100 | 50K/10K |
| Training data ratio on ImageNet | 100K/10K |
| Training epochs of the original large neural networks | 10 |
| Rewinding parameters to the training epochs of the original large neural networks | 10 |
| Batch size for computing the approximate Hessian Matrix | 128 |
| Batch size for training the neural networks | 100 |
| Learning rate | 0.1 |
| Number of training epochs | 160 |

### A.6 Potential Negative Societal Impacts and Limitations

In this work, all the three image datasets are open-released datasets [28, 10], which allow researchers to use for non-commercial research and educational purposes. All the three datasets are widely used in training/evaluating the image classification. All baseline codes are open-accessed resources that are from the GitHub and licensed under the MIT License, which only requires preservation of copyright and license notices and includes the permissions of commercial use, modification, distribution, and private use.

To our best knowledge, this work is the first to theoretically verify the Lottery Ticket Hypothesis (LTH) and the existence of winning lottery tickets by leveraging dynamical systems theory and inertial manifold theory. This work explores the possibility of theoretically lossless pruning as well as one-time pruning, compared with existing neural network pruning and LTH techniques. Our framework can be used in a wide variety of deep learning tasks in resource-intensive scenarios with the requirement of low latency and energy consumption, such as Internet of Things and mobile computing. This paper is primarily of a theoretical nature. We expect our findings to produce positive environmental impact, i.e, significantly improve the efficiency and scalability of deep learning models by reducing the time and space requirements of deep neural networks both at training and test time. To our best knowledge, we do not envision any immediate negative societal impacts of our results, such as security, privacy, and fairness issues.

An important product of this paper is to explore the possibility of theoretically lossless pruning as well as one-time pruning. Due to large size of neural networks in real scenarios as well as limit of current computing hardware, the approximate methods are utilized and designed to compute the Hessian matrix and estimate the Lipschitz constant for maintaining the efficiency. Our theoretical framework can inspire further improved development and implementations on neural network pruning with lossless pruning as well as remarkable efficiency from the academic institutions and industrial research labs.