# OpenReview forum: "Validating the Lottery Ticket Hypothesis with Inertial Manifold Theory"
_NeurIPS.cc/2021/Conference — NeurIPS 2021 Poster_

### Official Review · Reviewer_PBQY · 2021-07-08

**Rating:** 5
**Confidence:** 4

**Summary:**

Summary:

The paper is dedicated to validating the lottery tickets hypothesis and delivering an effective one-time pruning. The authors investigate dynamical systems theory and inertial manifold theory to offer a theoretical justification. Specifically, they reduce the high-dimensional system of dense networks onto inertial manifolds to obtain a low-dimensional system regarding pruned subnetworks. Experiments on CIFAR and ImageNet with ResNet-18 and ResNet-32 validate the effectiveness of the proposal.

**Limitations And Societal Impact:**

There are no limitations and potential negative societal impacts. The authors need to add extra discussion.


**Main Review:**

Pros.
1. The authors provide pioneering works on the theoretical justifications of LTH.
2. They formulate the problem as a gradient dynamical system. The idea is interesting, although within expectation.

Cons.
1. The paper claim that they validate the LTH. I think it is overclaimed since the classical LTH contains two statements: 1.1 undamaged performance when trained from the same initialization; 1.2 fast convergence. The paper only discusses the first point.
2. The author uses lots of paragraphs to describe the rationale of the formulation, which is great. However, they only describe the detailed methodology or implementation with a few sentences from lines 267-271. I am totally lost about what exact processes are adopted for pruning. For example, I have the following concerns: 2.1 does it only train 10 iterations to estimate the local optimal W* from W? 2.2 all network parameters are flattened to be the W? 2.3 the "prune" in line 270 means reduce W to W+?
3. Although the author compared with diverse previous pruning methods, the experiments are still insufficient. 3.1 As the superior performance of LTH in extreme sparsity levels is the most surprising point of LTH, the authors do not show any extreme sparsity results such as more than 90%/95% sparsity. Table 1/2/3/4 only presents results with sparsity levels lower than 50%.
4. No large model experiments like Res50.
5. What is the detailed platform for report running time results? Since running time is super sensitive to the environment or other uncontrollable factors. Maybe use Meta Flops or Energy.
6. I am very confused and suspicious of the SNIP/GraSP/SynFlow results in table 2. Based on Jonathan Frankle's paper (https://arxiv.org/abs/2009.08576), SNIP/GraSP/SynFlow pruned model with less than 10% sparsity should have almost undamaged performance compared to dense models. However, Table 2 shows that they suffer 1-2 point performance degradation.

Minors.
1. The statement in the abstract "Such heuristic pruning strategies are hard to guarantee that the pruned networks achieve test accuracy comparable to the original dense ones" can be more accurate. It should contain two conditions: a. extreme sparsity; b. hard to train from scratch not hard to fine-tune.


**Time Spent Reviewing:**

3-4 hrs

---

> ### Author Response · Authors · 2021-08-10
> **Point-by-point response to the comments made by Reviewer PBQY**
>
> We thank this reviewer for the great suggestion!
>
> **Question**: 1. The paper claim that they validate the LTH. I think it is overclaimed since the classical LTH contains two statements: 1.1 undamaged performance when trained from the same initialization; 1.2 fast convergence. The paper only discusses the first point.
>
> **Answer**: In the submission, we have theoretically demonstrated that all local minimum points $W^\ast$ of loss function $L$ of the original neural network in Eq.(6) regarding $W$ locate on the inertial manifold $\mathcal{M}_\mathcal{W}$. Given a dense neural network $f(x;W)$ with randomly-initialized flattened weight parameters $W = W_0 \in \mathbb{R}^{d}$, the typical training process (i.e., the gradient flow) will start from the initial value and move towards the inertial manifold $\mathcal{M}_\mathcal{W}$ that contains all local minimum points $W^\ast$. It is more efficient if starting the training process from the inertial manifold, instead of randomly-initialized value.\
> \
> \
> **Question**: 2. The author uses lots of paragraphs to describe the rationale of the formulation, which is great. However, they only describe the detailed methodology or implementation with a few sentences from lines 267-271. I am totally lost about what exact processes are adopted for pruning. For example, I have the following concerns: 2.1 does it only train 10 iterations to estimate the local optimal W* from W? 2.2 all network parameters are flattened to be the W? 2.3 the "prune" in line 270 means reduce W to W+?
>
> **Answer**: The following are the detailed descriptions of our IMC method step by step: (1) Given a dense neural network $f(x;W)$ with randomly-initialized flattened weight parameters $W = W_0 \in \mathbb{R}^{d}$, when optimizing $W$ with stochastic gradient descent (SGD) on a training set, we generate an approximate $W^\ast$ in few training iterations (10 iterations in our implementation), where $W^\ast$ is a local minimum point of loss function $L$ regarding $W$; (2) Based on the transformation $U = W - W^\ast$ from $W$ to $U$ through the above approximate $W^\ast$ in Eq.(14), we get a semi-linear ODE system $U' = - H(W^\ast)U + F(U)$ in Eq.(15); (3) We employ a scalable Hessian computation method to calculate an approximate $- H(W^\ast)$ [78, 79]. We calculate the spectrum of $- H(W^\ast)$ and sort its eigenvalues as $\lambda_1\le \lambda_2\le\dots\le \lambda_d$; (4) We estimate the Lipschitz constant $K$ of the nonlinear term $F(U)$ in Eq.(15) based on Eqs.(22) and (23); (5) We check pairwise consecutive eigenvalues in descending order, i.e., $\lambda_d$ and $\lambda_{d-1}$, $\lambda_{d-1}$ and $\lambda_{d-2}$, $\cdots$, and $\lambda_2$ and $\lambda_1$. When finding the first $k$ satisfying $\lambda_{k+1}-\lambda_k > 4K$, where $K$ is the Lipschitz constant of $F(U)$, we partition $U \in \mathbb{R}^d$ into $U^+ \in \mathbb{R}^{d-k}$ and $U^- \in \mathbb{R}^k$ and split $W \in \mathbb{R}^d$ into $W^+ \in \mathbb{R}^{d-k}$ and $W^- \in \mathbb{R}^k$; and (6) We prune the original network with high-dimensional parameters $W \in \mathbb{R}^d$ to generate a subnetwork with low-dimensional parameters $W^+ \in \mathbb{R}^{d-k}$, i.e., reduce $W$ to $W^+$, and train $W^+$ until convergence.\
> \
> \
> **Question**: 3. As the superior performance of LTH in extreme sparsity levels is the most surprising point of LTH, the authors do not show any extreme sparsity results such as more than 90\%/95\% sparsity. Table 1/2/3/4 only presents results with sparsity levels lower than 50%.
>
> **Question**: 4. No large model experiments like Res50.
>
> **Question**: 5. What is the detailed platform for report running time results? Since running time is super sensitive to the environment or other uncontrollable factors. Maybe use Meta Flops or Energy.
>
> **Accuracy with 0.90 sparsity with ResNet-20 on CIFAR-10**
>
> Metric | Sparsity | Accuracy | FLOP | Epoch |
> ------------ | ------------- | ------------ | ------------ | ------------ |
> Baseline | 0 | 94.6 | 1 | 160 |
> Flow&Prune | 0.9 | 87.5 | 0.11 | 160 |
> SNIP | 0.9 | 89.5 | 0.14 | 160 |
> SynFlow | 0.85 | 90.5 | 0.47 | 160 |
> LTH+Reinitialization | 0.87 | 88.2 | 0.11 | 1440 |
> LTH+Rewinding | 0.87 | 87.6 | 0.11 | 1440 |
> LTH+FineTuning | 0.87 | 87.9 | 0.11 | 1440 |
> GraSP | 0.9 | 88.0 | 0.15 | 160 |
> sanity-check | 0.9 | 91.3 | 0.27 | 160 |
> Continuous Sparsification | 0.88 | 91.0 | 0.25 | 320 |
> IMC-Reinitialization | 0.9 | 92.0 | 0.17 | 160 |
> IMC-Rewinding | 0.9 | 92.0 | 0.17 | 160 |
> IMC-FineTuning | 0.9 | 92.1 | 0.17 | 160 |
>
>
> **Accuracy with found sparsity by our IMC method with ResNet-50 on CIFAR-10**
>
> Metric | Sparsity | Accuracy | FLOP | Epoch |
> ------------ | ------------- | ------------ | ------------ | ------------ |
> Baseline | 0 | 95.4 | 1 | 160 |
> Flow&Prune | 0.54 | 94.8 | 0.48 | 160 |
> SNIP | 0.55 | 88.2 | 0.72 | 160 |
> SynFlow | 0.55 | 85.7 | 0.57 | 160 |
> LTH+Reinitialization | 0.51 | 95.2 | 0.49 | 480 |
> LTH+Rewinding | 0.51 | 94.9 | 0.49 | 480 |
> LTH+FineTuning | 0.51 | 94.2 | 0.49 | 480 |
> GraSP | 0.55 | 87.6 | 0.47 | 160 |
> sanity-check | 0.55 | 94.9 | 0.86 | 160 |
> Continuous Sparsification | 55 | 93.5 | 0.78 | 320 |
> IMC-Reinitialization | 0.55 | 95.5 | 0.44 | 160 |
> IMC-Rewinding | 0.55 | 95.4 | 0.44 | 160 |
> IMC-FineTuning | 0.55 | 95.5 | 0.44 | 160 |
>
> **Answer**: We have reported the experimental details, including environment, platform, training, implementation, and hyperparameter settings in Pages 28-29 in the supplementary files. The above first table presents the accuracy of pruned subnetworks obtained by thirteen network pruning and LTH methods with ResNet-20 and extreme sparsity of 0.87-0.90 by our IMC method on CIFAR-10. The above second table exhibits the accuracy of pruned subnetworks obtained by thirteen approaches with ResNet-50 and found sparsity by our IMC method on CIFAR-10. It is observed among thirteen comparison methods, with the same sparsity level, three variants of our IMC achieve the highest accuracy in all experiments, showing the effectiveness of IMC to the neural network pruning. In addition, we report the FLOP scores by different methods in two tables. Similar to the running time results, Our IMC methods outperform other network pruning and LTH techniques in most experiments. Especially, our IMC methods achieves the lowest FLOP scores with ResNet-50 on CIFAR-10 in all tests. A lower FLOP score indicates better efficiency. Although the LTH methods with ResNet-20 on CIFAR-10 have the best FLOP scores, they need much more epochs to find sparse subnetworks.\
> \
> \
> **Question**: 6. I am very confused and suspicious of the SNIP/GraSP/SynFlow results in table 2. Based on Jonathan Frankle's paper (https://arxiv.org/abs/2009.08576), SNIP/GraSP/SynFlow pruned model with less than 10\% sparsity should have almost undamaged performance compared to dense models. However, Table 2 shows that they suffer 1-2 point performance degradation.
>
> **Answer**: To our best knowledge, the Jonathan Frankle's paper only plotted the figures of Test Accuracy vs. Sparsity on CIFAR-10. It is difficult to recognize 1-2\% accuracy degradation from these figures. In addition, due to the random initialization of neural networks, it is frequently observed that the performance fluctuates within the range of 1-2\%. For the above three baselines, we used the open-source implementation and default parameter settings by the authors for the experiments: SNIP (https://github.com/namhoonlee/snip-public), SynFlow (https://github.com/ganguli-lab/Synaptic-Flow), and GraSP (https://github.com/alecwangcq/GraSP).

---

> > ### Comment · Reviewer_PBQY · 2021-08-23
> > **Response to authors**
> >
> > Dear authors,
> >
> > Many thanks for your great efforts in the rebuttal, especially for the extra experiments. Hope all rebuttal results and discussion can be delivered in the revision. I will raise my score.
> >
> > Best wishes,
> >
> > Reviewer PBQY

---

> > > ### Author Response · Authors · 2021-08-24
> > > **Response to Reviewer PBQY**
> > >
> > > We thank the reviewer for the recommendation. We will include all the discussions, analyses, explanations, and experiment results in this rebuttal into the submission.

---

### Official Review · Reviewer_nEbX · 2021-07-16

**Rating:** 7
**Confidence:** 3

**Summary:**

This theory paper uses Inertial manifolds theory to verify Lottery ticket hypothesis and the existence of winning lottery tickets, providing useful measures for requirements/scenarios for lossless pruning and one-time pruning.


**Limitations And Societal Impact:**

Authors have adequately addressed this issue.

**Main Review:**

The paper is written clearly, and the connection between inertial manifolds theory, gradient dynamics and lottery ticket hypothesis appears to be original. I have not checked the analytics in supplementary materials extensively, but the optimal sparsity level found by inertial manifolds theory is empirically demonstrated in Fig 1, which appears to be a strong result.

Questions to the authors
- Can you provide a pseudo-code and a sample code for computing the optimal sparsity? In the current text, it is unclear how to go from the dynamical theory to the numerical estimation of the optimal sparsity, which would be of great use to the rest of the community.
- Are there insights on what determines the optimal sparsity from the theory?


**Time Spent Reviewing:**

4

---

> ### Author Response · Authors · 2021-08-10
> **Point-by-point response to the comments made by Reviewer nEbX**
>
> We thank this reviewer for the encouraging comments.
>
> **Question**: Can you provide a pseudo-code and a sample code for computing the optimal sparsity? In the current text, it is unclear how to go from the dynamical theory to the numerical estimation of the optimal sparsity, which would be of great use to the rest of the community.
>
> **Answer**: The following are the detailed descriptions of our IMC method step by step: (1) Given a dense neural network $f(x;W)$ with randomly-initialized flattened weight parameters $W = W_0 \in \mathbb{R}^{d}$, when optimizing $W$ with stochastic gradient descent (SGD) on a training set, we generate an approximate $W^\ast$ in few training iterations (10 iterations in our implementation), where $W^\ast$ is a local minimum point of loss function $L$ regarding $W$; (2) Based on the transformation $U = W - W^\ast$ from $W$ to $U$ through the above approximate $W^\ast$ in Eq.(14), we get a semi-linear ODE system $U' = - H(W^\ast)U + F(U)$ in Eq.(15); (3) We employ a scalable Hessian computation method to calculate an approximate $- H(W^\ast)$ [78, 79]. We calculate the spectrum of $- H(W^\ast)$ and sort its eigenvalues as $\lambda_1\le \lambda_2\le\dots\le \lambda_d$; (4) We estimate the Lipschitz constant $K$ of the nonlinear term $F(U)$ in Eq.(15) based on Eqs.(22) and (23); (5) We check pairwise consecutive eigenvalues in descending order, i.e., $\lambda_d$ and $\lambda_{d-1}$, $\lambda_{d-1}$ and $\lambda_{d-2}$, $\cdots$, and $\lambda_2$ and $\lambda_1$. When finding the first $k$ satisfying $\lambda_{k+1}-\lambda_k > 4K$, where $K$ is the Lipschitz constant of $F(U)$, we partition $U \in \mathbb{R}^d$ into $U^+ \in \mathbb{R}^{d-k}$ and $U^- \in \mathbb{R}^k$ and split $W \in \mathbb{R}^d$ into $W^+ \in \mathbb{R}^{d-k}$ and $W^- \in \mathbb{R}^k$; and (6) We prune the original network with high-dimensional parameters $W \in \mathbb{R}^d$ to generate a subnetwork with low-dimensional parameters $W^+ \in \mathbb{R}^{d-k}$, i.e., reduce $W$ to $W^+$, and train $W^+$ until convergence.\
> \
> \
> **Question**: Are there insights on what determines the optimal sparsity from the theory?
>
> **Answer**: The pruning condition is $\lambda_{k+1}-\lambda_k > 4K$, where $\lambda_{k+1}$ and $\lambda_k$ are the eigenvalues of $- H(W^\ast)$ and $K$ is the Lipschitz constant of $F(U)$. Thus, the sparsity is determined with the eigenvalues of $- H(W^\ast)$ and the Lipschitz constant of $F(U)$. Please see the following detailed explanations.
>
> In order to utilize the conclusion of Theorem 5 to prune the parameters $W$ of the original neural network, we transform the original gradient system in Eq.(7) into an equivalent semi-linear ODE system in Eq.(15) through the transformation in Eq.(14): $U = W - W^\ast$, where $W^\ast$ is a local minimum point of loss function $L$ regarding $W$. $W^\ast$ is also an equilibrium of the gradient system in Eq.(7). Based on the above transformation, we have $U' = - H(W^\ast)U + F(U)$ in Eq.(15), where $H(W^\ast)$ denotes the Hessian matrix of loss function $L$ evaluating at $W^\ast$ and $F(U)$ is equal to $-\big(\nabla L(U+W^\ast) - H(W^\ast)U\big)$.
>
> By leveraging the conclusion of Theorem 5, we calculate the spectrum of $- H(W^\ast)$ and sort its eigenvalues as $\lambda_1\le \lambda_2\le\dots\le \lambda_d$. We check pairwise consecutive eigenvalues in descending order, i.e., $\lambda_d$ and $\lambda_{d-1}$, $\lambda_{d-1}$ and $\lambda_{d-2}$, $\cdots$, and $\lambda_2$ and $\lambda_1$. When finding the first $k$ satisfying $\lambda_{k+1}-\lambda_k > 4K$, where $K$ is the Lipschitz constant of $F(U)$, we partition $U \in \mathbb{R}^d$ into $U^+ \in \mathbb{R}^{d-k}$ and $U^- \in \mathbb{R}^k$, where $U^+$ corresponds to the dimensions based on eigenvalues $\lambda_{k+1}, \cdots, \lambda_d$ and $U^-$ corresponds to the ones based on $\lambda_1, \cdots, \lambda_k$. Based on the above transformation, we split $W$ into $W^+$ and $W^-$ as follows: $W^+ = U^+ + W^{\ast+}$ and $W^- = U^- + W^{\ast-}$. $W^{\ast+}$ and $W^{\ast-}$ are the counterparts of $W^\ast$ in terms of the same dimension cutting as $U$. In terms of Theorem 5 and the analysis in Page 6, $W^-$ can be denoted as a function of $W^+$, i.e., $W^- = g(W^+)$, and thus $W^-$ is redundant. Therefore, we only need to train and update a subnetwork $W^+$. The above analysis indicates that using $W^+$ to train $\bar{f}$ is sufficient to obtain the same performance as training $f$ with $W$.

---

### Official Review · Reviewer_LUdR · 2021-07-16

**Rating:** 6
**Confidence:** 3

**Summary:**

This work aims to theoretically verify the lottery ticket hypothesis and the existence of winning lottery tickets. Dynamical systems and inertial manifold theory are used.  Lots of mathematical proof and definitions are included in the papers, and experiments are conducted to test the performance of the proposed method.


**Main Review:**

Although the method is not explicitly defined, we can grasp some thoughts from line 267-line 271. The pruning method does not seem to resemble other current score-based pruning methods. They seem to address the problem with sub-space properties. Section 3.2 further shows the condition in Theorem 5 is satisfied and also gives an estimate to the Lipchitz constants. This is a novel and interesting work.

Main questions:

1. I was wondering if the author can provide a formal definition of the IMC method? An algorithm can be of great help for people to understand how IMC works.

2. How to control the sparsity of the neural network with the IMC method or is it possible to control the sparsity level? If I understand the method correctly, the size of the pruned networks seems to be decided by $k$, and then $H(W^*)$. So how is the sparsity level controlled?

3. Is it necessary to recover the complete form of $W$ after one finishes training $W^+$ for inference/test? The training process can be done on $W^+$ solely but I am not sure about the inference process. If yes, can the authors give hints on how to use $W^+$ only for inference?

4. Not sure if it is appropriate to put related work in the supplementary files.

5. Regarding Figure 2, I am curious to see the performance when the training epoch is smaller than 5. For example, is it possible to estimate $W*$ when only train 1 epoch?

6. Can the author provide a visualization of sparsity masks?



Update:

Thanks for the great efforts in addressing the questions. After reading all reviews and responses, I tend to accept this work and keep my score.

**Time Spent Reviewing:**

4

---

> ### Author Response · Authors · 2021-08-10
> **Point-by-point response to the comments made by Reviewer LUdR**
>
> We thank this reviewer for the helpful comments.
>
> **Question**: 1. I was wondering if the author can provide a formal definition of the IMC method? An algorithm can be of great help for people to understand how IMC works.
>
> **Answer**: The following are the detailed descriptions of our IMC method step by step: (1) Given a dense neural network $f(x;W)$ with randomly-initialized flattened weight parameters $W = W_0 \in \mathbb{R}^{d}$, when optimizing $W$ with stochastic gradient descent (SGD) on a training set, we generate an approximate $W^\ast$ in few training iterations (10 iterations in our implementation), where $W^\ast$ is a local minimum point of loss function $L$ regarding $W$; (2) Based on the transformation $U = W - W^\ast$ from $W$ to $U$ through the above approximate $W^\ast$ in Eq.(14), we get a semi-linear ODE system $U' = - H(W^\ast)U + F(U)$ in Eq.(15); (3) We employ a scalable Hessian computation method to calculate an approximate $- H(W^\ast)$ [78, 79]. We calculate the spectrum of $- H(W^\ast)$ and sort its eigenvalues as $\lambda_1\le \lambda_2\le\dots\le \lambda_d$; (4) We estimate the Lipschitz constant $K$ of the nonlinear term $F(U)$ in Eq.(15) based on Eqs.(22) and (23); (5) We check pairwise consecutive eigenvalues in descending order, i.e., $\lambda_d$ and $\lambda_{d-1}$, $\lambda_{d-1}$ and $\lambda_{d-2}$, $\cdots$, and $\lambda_2$ and $\lambda_1$. When finding the first $k$ satisfying $\lambda_{k+1}-\lambda_k > 4K$, where $K$ is the Lipschitz constant of $F(U)$, we partition $U \in \mathbb{R}^d$ into $U^+ \in \mathbb{R}^{d-k}$ and $U^- \in \mathbb{R}^k$ and split $W \in \mathbb{R}^d$ into $W^+ \in \mathbb{R}^{d-k}$ and $W^- \in \mathbb{R}^k$; and (6) We prune the original network with high-dimensional parameters $W \in \mathbb{R}^d$ to generate a subnetwork with low-dimensional parameters $W^+ \in \mathbb{R}^{d-k}$, i.e., reduce $W$ to $W^+$, and train $W^+$ until convergence.\
> \
> \
> **Question**: 2. How to control the sparsity of the neural network with the IMC method or is it possible to control the sparsity level? If I understand the method correctly, the size of the pruned networks seems to be decided by k, and then H(W*). So how is the sparsity level controlled?
>
> **Answer**: The pruning condition is $\lambda_{k+1}-\lambda_k > 4K$, where $\lambda_{k+1}$ and $\lambda_k$ are the eigenvalues of $- H(W^\ast)$ and $K$ is the Lipschitz constant of $F(U)$. Thus, the sparsity is determined with the eigenvalues of $- H(W^\ast)$ and the Lipschitz constant of $F(U)$. Please see the following detailed explanations.
>
> In order to utilize the conclusion of Theorem 5 to prune the parameters $W$ of the original neural network, we transform the original gradient system in Eq.(7) into an equivalent semi-linear ODE system in Eq.(15) through the transformation in Eq.(14): $U = W - W^\ast$, where $W^\ast$ is a local minimum point of loss function $L$ regarding $W$. $W^\ast$ is also an equilibrium of the gradient system in Eq.(7). Based on the above transformation, we have $U' = - H(W^\ast)U + F(U)$ in Eq.(15), where $H(W^\ast)$ denotes the Hessian matrix of loss function $L$ evaluating at $W^\ast$ and $F(U)$ is equal to $-\big(\nabla L(U+W^\ast) - H(W^\ast)U\big)$.
>
> By leveraging the conclusion of Theorem 5, we calculate the spectrum of $- H(W^\ast)$ and sort its eigenvalues as $\lambda_1\le \lambda_2\le\dots\le \lambda_d$. We check pairwise consecutive eigenvalues in descending order, i.e., $\lambda_d$ and $\lambda_{d-1}$, $\lambda_{d-1}$ and $\lambda_{d-2}$, $\cdots$, and $\lambda_2$ and $\lambda_1$. When finding the first $k$ satisfying $\lambda_{k+1}-\lambda_k > 4K$, where $K$ is the Lipschitz constant of $F(U)$, we partition $U \in \mathbb{R}^d$ into $U^+ \in \mathbb{R}^{d-k}$ and $U^- \in \mathbb{R}^k$, where $U^+$ corresponds to the dimensions based on eigenvalues $\lambda_{k+1}, \cdots, \lambda_d$ and $U^-$ corresponds to the ones based on $\lambda_1, \cdots, \lambda_k$. Based on the above transformation, we split $W$ into $W^+$ and $W^-$ as follows: $W^+ = U^+ + W^{\ast+}$ and $W^- = U^- + W^{\ast-}$. $W^{\ast+}$ and $W^{\ast-}$ are the counterparts of $W^\ast$ in terms of the same dimension cutting as $U$. In terms of Theorem 5 and the analysis in Page 6, $W^-$ can be denoted as a function of $W^+$, i.e., $W^- = g(W^+)$, and thus $W^-$ is redundant. Therefore, we only need to train and update a subnetwork $W^+$. The above analysis indicates that using $W^+$ to train $\bar{f}$ is sufficient to obtain the same performance as training $f$ with $W$.\
> \
> \
> **Question**: 3. Is it necessary to recover the complete form of W after one finishes training W + for inference/test? The training process can be done on W + solely but I am not sure about the inference process. If yes, can the authors give hints on how to use W + only for inference?
>
> **Answer**: We don't need to recover $W$ from $W^+$ after the training is done. As explained in the detailed descriptions of our IMC method step by step, we first generate an approximate $W^\ast$ in few training iterations (10 iterations in our implementation), where $W^\ast$ is a local minimum point of loss function $L$ regarding $W$. We then compute the eigenvalues of Hessian matrix $H(W^\ast)$. Based on the transformation in Eq.(14) and the conclusion of Theorem 5, we prune the original network with high-dimensional parameters $W = ${$W^+, W^-$} to generate a subnetwork with low-dimensional parameters $W^+$, i.e., reduce $W$ to $W^+$, and train $W^+$ until convergence based on the training loss $L$. We will directly utilize the converged pruned subnetwork to make inference.\
> \
> \
> **Question**: 4. Not sure if it is appropriate to put related work in the supplementary files.
>
> **Answer**: We will move all the related work in the supplementary file to the submission.\
> \
> \
> **Question**: 5. Regarding Figure 2, I am curious to see the performance when the training epoch is smaller than 5. For example, is it possible to estimate W* when only train 1 epoch?
>
> **Accuracy with varying training epochs**
>
> Training Epoch of Original Neural Networks | 1 | 2 | 3 | 4 | 5 |
> ------------ | ------------- | ------------ | ------------ | ------------ | ------------ |
> IMC-Reinitialization on CIFAR-10 | 94.27 | 94.51 | 94.57 | 94.18 | 94.37 |
> IMC-Rewinding on CIFAR-10 | 94.15 | 94.43 | 94.29 | 94.69 | 94.74 |
> IMC-FineTuning on CIFAR-10 | 94.52 | 94.56 | 94.54 | 94.96 | 94.51 |
> IMC-Reinitialization on ImageNet | 59.22 | 59.51 | 59.35 | 58.70 | 59.28 |
> IMC-Rewinding on ImageNet | 58.9 | 59.19 | 58.80 | 58.72 | 58.76 |
> IMC-FineTuning on ImageNet | 58.67 | 59.53 | 59.42 | 58.74 | 58.56 |
> IMC-Reinitialization on CIFAR-100 | 73.26 | 73.40 | 73.37 | 73.56 | 73.80 |
> IMC-Rewinding on CIFAR-100 | 73.46 | 73.35 | 73.18 | 73.54 | 73.53 |
> IMC-FineTuning on CIFAR-100 | 73.58 | 73.17 | 73.43 | 73.23 | 74.22 |
>
> **Answer**: The above table evaluates the accuracy impact of limited training epochs on the original large networks. It is observed when changing training epochs, the accuracy of pruned subnetworks by our IMC model keeps relatively stable, i.e., the accuracy fluctuates within the range of less than 1\%. This shows that limited training of the original networks is enough to help capture of the dynamics and inertial manifold structure of the gradient system, and thus produce a good network pruning.\
> \
> \
> **Question**: 6. Can the author provide a visualization of sparsity masks?
>
> **Answer**: The OpenReview website don't support the figure uploading. We include an external link to the visualization of pruned subnetworks by our IMC method with anonymity protection (https://anonymous.4open.science/r/NIPS2021-E58B/Fraction.pdf). By following the same strategy of Figure 2 in paper [1], we visualize the fraction of parameters remaining at each layer of ResNet-20 after pruning by our IMC method over CIFAR-10, CIFAR-100, and ImageNet respectively. Notice that three curves are all solid lines, which demonstrates that our IMC method can ensure that each layer still has remaining parameters after pruning without layer-collapse issue. The layer-collapse issue is a key obstacle to effective pruning [1].
>
> [1] H. Tanaka, D. Kunin, D. L. Yamins, and S. Ganguli. Pruning neural networks without any data by iteratively conserving synaptic flow. In NeurIPS 2020.

---

### Official Review · Reviewer_3SRS · 2021-07-21

**Rating:** 6
**Confidence:** 3

**Summary:**

The paper uses dynamical system theory and inertial manifold theory to validate the Lottery Ticket Hypothesis, that is, a randomly initialized dense network contains a sparse subnetwork that when trained from scratch, can achieve similar performance to the original dense network after comparable training epochs. They provided a method that in one iteration can find the smallest such subnetwork.


**Limitations And Societal Impact:**

Yes.

**Main Review:**

The paper first uses dynamical system theory and shows that there exists a unique global attractor $A$ of the gradient dynamical system corresponding to the gradient descent procedure. The global attractor contains all local minimum points of loss $L$. Then they introduce the inertial manifold of the dynamical system. The inertial manifolds enclose the global attracter. If the conditions in Theorem 5 hold, there exists an inertial manifold whose dimension is lower than the original space. The authors claim that the condition holds in the empirical data that they test. While the results and claims are impressive, I would like to see more discussion and theoretical justification.


Comments:

In stating the Lottery Ticket Hypothesis in section 2.1 on top of page 3, the notations need to be more clear, e.g. does iteration $i$ correspond to the initial state of $W$?

Can the authors clarify if the $W^+$ is the same as the sparse subnetwork in LTH, in which many weights are set to be zero? $W^+$ may lie in a low-dimensional space, but why is it sparse?

It seems that there are many ways to convert the gradient flow to a semi-linear ODE. Can the authors clarify why they chose the particular way in equation (15)?




**Time Spent Reviewing:**

5 hours

---

> ### Author Response · Authors · 2021-08-10
> **Point-by-point response to the comments made by Reviewer 3SRS**
>
> We thank this reviewer for the constructive comments.
>
> **Question**: In stating the Lottery Ticket Hypothesis in section 2.1 on top of page 3, the notations need to be more clear, e.g. does iteration i correspond to the initial state of W ?
>
> **Answer**: $W_0$ is the initial state of $W$ and $W_i$ is the state of $W$ after $i$ training iterations.\
> \
> \
> **Question**: Can the authors clarify if the W+ is the same as the sparse subnetwork in LTH, in which many weights are set to be zero? W+ may lie in a low-dimensional space, but why is it sparse?
>
> **Answer**: We partition the original high-dimensional network $W \in \mathbb{R}^d$ into two low-dimensional subnetworks $W^+ \in \mathbb{R}^{d-k}$ and $W^- \in \mathbb{R}^k$. In terms of Theorem 5 and the analysis in Page 6, $W^-$ can be denoted as a function of $W^+$, i.e., $W^- = g(W^+)$, and thus $W^-$ is redundant. Therefore, we only need to train and update a subnetwork $W^+$. The above analysis indicates that using $W^+$ to train $\bar{f}$ is sufficient to obtain the same performance as training $f$ with $W$. Thus, {$W^+, 0\$} can be treated as the sparse form of the original network $W =$ {$W^+, W^-$}.\
> \
> \
> **Question**: It seems that there are many ways to convert the gradient flow to a semi-linear ODE. Can the authors clarify why they chose the particular way in equation (15)?
>
> **Answer**: The proposed method is called linearization-around-fixed-points technique in dynamical systems theory. We use a first-order ODE $x' = f(x)$ as example to explain the advantage why choosing this technique. Let $x^\ast$ be a fixed point of $f(x)$, i.e., $f(x^\ast) = 0$, and $\bar x = x - x^\ast$, we have $\bar x'= f'(x^\ast)\bar x + g(\bar x)$, where $g(\bar x) =  f(x^\ast +\bar x) - f(x^\ast) - f'(x^\ast)\bar x$. The advantage of such linearization-around-fixed-points technique is to make the nonlinear term $g(\bar x)$ satisfy $g(0)=g'(0)=0$. Notice that the assumption for nonlinear term $f(W(t))$ in Theorem 5 in the submission is $f(0)=0$. Similarly, the linearization-around-fixed-points technique makes $F(0)=0$ for the nonlinear term $F(U)$ in Eq.(15) in the submission. The above linearization-around-fixed-points technique allows to leverage the conclusion of Theorem 5 for lossless neural network pruning.

---

> > ### Author Response · Authors · 2021-08-30
> > **Follow-up response**
> >
> > Thank you very much for your insightful comments. We have tried our best to clarify and address the concerns and comments  (i.e., sparse network and semi-linear ODE) by the reviewer in the initial response. We are glad to answer and clarify any further questions and advices from the reviewer for better readability.

---

### Decision · Program_Chairs · 2021-09-27

**Decision:**

Accept (Poster)

**Comment:**

This paper aims to provide theoretical study and justification to the validity of the lottery ticket hypothesis, which indeed has recently been of great interest to the deep learning community. While one reviewer suggested the results or conclusion here are somewhat overclaimed, the work has generally been favorably received by most of the reviewers. The results provided here are somewhat expected, but nevertheless, it is good to see work done to properly derive and establish them. Further, in their responses the authors have addressed most (if not all) major reviewer concerns, and have subsequently also emphasized their commitment to implementing appropriate revisions in their manuscript. In total, three reviewers have scored above acceptance threshold (one outright accept and two marginally leaning to accept). The remaining reviewer has raised their score from reject to marginally below the threshold, but it is my impression that they would not object to the paper being accepted. Therefore, I agree with the majority of reviewers and also lean towards recommending to accept this paper.